# Metabolic costs and trade-offs of hypermetabolism in human motor neurons with ATP synthase deficiency
Rubén Torregrosa-Muñumer [1] ✉, Jeremi Turkia [1], Rumeysa Ermiş[1], Jouni Kvist [1], Sandra Harjuhaahto[1], Jana Pennonen[1], Ville Hietakangas [2], Emil Ylikallio[1,3] & Henna Tyynismaa [1] ✉

Hypermetabolism, a futile cycle of energy production and consumption, has been proposed as an adaptive response to deficiencies in mitochondrial oxidative phosphorylation. However, the cellular costs of hypermetabolism remain largely unknown. Here we studied the consequences of hypermetabolism in human motor neurons harboring a heteroplasmic mutation in *MT-ATP6*, which impairs ATP synthase assembly. Respirometry, metabolomics, and proteomics analyses of the motor neurons showed that elevated ATP production rates were accompanied with increased demand for acetyl-Coenzyme A (acetyl-CoA) and depleted pantothenate (vitamin B5), and the proteome was remodeled to support the metabolic adaptation. Mitochondrial membrane potential and coupling efficiency remained stable, and the therapeutic agent avanafil did not affect metabolite levels. However, a redistribution of acetyl-CoA usage resulted in metabolic trade-offs, including reduced histone acetylation and altered maintenance of the neurotransmitter acetylcholine, revealing potential vulnerabilities in motor neurons. These findings advance the understanding of cellular metabolic consequences imposed by hypermetabolic conditions.

All living cells have evolved to maintain a delicate equilibrium between energy production and consumption, finely tuned to meet physiological demands while avoiding metabolic stress. Over evolution, this energy economy has been refined to impose boundaries that prevent metabolic overload during fluctuating conditions[1,2]. This balance is fundamental to cellular and organismal health, as both insufficient and excessive energy expenditure may compromise function and viability[3]. In eukaryotic cells, ATP, the primary cellular energy currency, is generated through two main metabolic pathways: glycolysis in the cytosol and oxidative phosphorylation (OxPhos) in the mitochondria. While glycolysis generates modest amounts of ATP from glucose, OxPhos uses redox equivalents from nutrient catabolism to power the electron transport chain (ETC), which creates a proton gradient across the inner mitochondrial membrane. This proton motive force fuels the ATP synthase (Complex V), a large enzyme complex that produces substantial amounts of ATP by coupling proton reentry with ATP synthesis.

Deficiency in OxPhos, caused by disease variants in mitochondrial DNA (mtDNA) or nuclear genes for mitochondrial proteins, commonly leads to reduced mitochondrial ATP production and a subsequent energy deficit in cells[4,5]. Cells can adapt to the energy deficits through different mechanisms. If OxPhos is impaired, cells may shift towards glycolysis to generate ATP[6]. Although less efficient, glycolysis may still produce enough ATP to support cellular function. The metabolic shift also results in the accumulation of lactate as a byproduct, which can lead to lactic acidosis, a common hallmark of impaired OxPhos function. Increased ATP demand may also be partially compensated by upregulating mitochondrial biogenesis, with the goal of improving mitochondrial efficiency[7]. However, generating new mitochondria also imposes a significant resource and energy burden on the cell. Occasionally, the balance is disrupted not by energy deficit but by pathological increases in energy turnover. In rare syndromes such as Hypermetabolism due to Uncoupled Mitochondrial Oxidative Phosphorylation 1 (HUMOP1)[8–10] and 2 (HUMOP2)[11,12], uncoupling of the proton motive force from ATP synthesis paradoxically results in increased OxPhos activity. HUMOP1, also known as Luft syndrome, has an unknown genetic basis, whereas HUMOP2 is caused by a dominant variant in the nuclear gene *ATP5F1B*, which encodes a subunit of the F$_1$ domain of the ATP synthase. Demonstrating the effects of increased metabolism on systemic level, these rare hypermetabolic conditions led to recurrent hyperthermia and weight loss or an inability to gain weight despite excessive caloric intake.

[1]Stem Cells and Metabolism Research Program, Faculty of Medicine, University of Helsinki, Helsinki, Finland. [2]Faculty of Biological and Environmental Sciences & Institute of Biotechnology, University of Helsinki, Helsinki, Finland. [3]Clinical Neurosciences, Neurology, Helsinki University Hospital, Helsinki, Finland. ✉e-mail: ruben.torregrosa@helsinki.fi; henna.tyynismaa@helsinki.fi

Hypermetabolism, which can be defined as an increase in resting energy expenditure (REE) of as little as 10% above normal levels[13–17], reflects a state of elevated energy expenditure within cells. This increase can arise from increased ATP turnover, indicating elevated ATP demand (from biosynthetic, ionic and transport, or contractile processes[18]), or from increased uncoupling of OxPhos, as seen in conditions such as Luft's disease[8–10]. While such mechanisms may initially compensate for the energy deficit, ultimately represents an unsustainable metabolic state that can exacerbate cellular dysfunction and disease manifestations over time. This chronic overload can trigger stress responses, alter cytokine and metabokine profiles, and reshape the epigenetic landscape[14]. However, knowledge of the specific molecular mechanisms of hypermetabolism within mitochondria and its consequences for the cell is lacking. Moreover, cell type specific metabolic adaptations to hypermetabolism, especially in post-mitotic cells with high energy demands such as neurons, are poorly known.

In this study, we investigated the mechanisms driving hypermetabolism and its metabolic impacts in human induced pluripotent stem cell-derived motor neurons (iPSC-MN)[19], in a controlled isogenic setting, combining respirometry, metabolomics, and proteomics. We found that elevated primary biosynthetic and bioenergetic pathways, including glycolysis and OxPhos, led to a higher demand of the hub metabolite acetyl-CoA, as evidenced by its increased biosynthetic machinery and depletion of its precursor, pantothenate (vitamin B5). Moreover, we observed a potential shift in the prioritization of acetyl-CoA usage resulting in trade-offs, including histone acetylation, which contributes to recalibration of the epigenetic regulation, and a potential vulnerability in maintaining acetylcholine (ACh) neurotransmitter levels in motor neurons. These findings provide new insights into the cell-type-specific secondary changes induced by hypermetabolism, potentially opening new avenues for understanding and devising treatments for metabolic and motor neuron diseases.

## Results

### 50% *MT-ATP6* heteroplasmy does not affect motor neuron differentiation but leads to hypermetabolism

Earlier studies of hypermetabolism mainly used proliferative cells such as patient-derived fibroblasts[11,14,15], which do not fully recapitulate the phenotypes of commonly affected post-mitotic cells, including neurons. We previously observed a paradoxical increase in mitochondrial respiration along with elevated lactate levels, indicative of a hypermetabolic state, in a patient-specific iPSC-MN model of peripheral neuropathy[19]. The studied motor neurons carried a 49% heteroplasmic nonsense variant (mitochondrial mutation load) in *MT-ATP6* (m.9154 C > T, p.Gln210Ter), which encodes the "a" subunit of the $F_0$ domain of the ATP synthase, forming the proton re-entry channel[20]. Higher heteroplasmy levels (~70%) of this *MT-ATP6* variant hampered cellular reprogramming and motor neuron differentiation[19], indicating the severity of the variant on cellular level. Disease variants in *MT-ATP6* typically lead to neurodegeneration, but the pathogenic mechanisms and disease manifestations vary widely[21–23]. To further investigate the neuronal hypermetabolism in the *MT-ATP6* neurons, we expanded our research to include two additional mutant iPSC clones with approximately 50% heteroplasmy (52% and 54%) and two isogenic controls (both with 0% heteroplasmy, (1) and (2)) (Fig. 1A).

We first tested whether our previous findings of the 49% mutant were valid also in the iPSC clones with 52% and 54% heteroplasmy of the variant. The *MT-ATP6* nonsense variant disrupted the assembly of ATP synthase (complex V) in iPSC in proportion to the heteroplasmy levels, as indicated by an increase in subcomplexes (sc) (Fig. 1B). Yet, using a cocktail of small molecules we successfully differentiated the mutant iPSC into spinal motor neurons (Supplementary Fig. 1A), with comparable numbers of ISL1/2- and Hb9-positive cells (Fig. 1C–E). We did not observe any significant differences in mtDNA copy number in iPSC-MN (Supplementary Fig. 1B), indicating that the mitochondrial mass did not change, and the mutation heteroplasmy load remained comparable throughout the differentiation (Supplementary Fig. 1C, D).

We used a Seahorse XF analyzer normalized to total DNA (proxy of cell number) to measure the oxygen consumption rate (OCR) and found that motor neurons with ~50% heteroplasmy exhibited increased mitochondrial respiration, or OxPhos, parameters, including basal mitochondrial respiration, proton leak and maximal respiration (Fig. 1F, G, and Supplementary Fig. 1F). However, coupling efficiency and % spare capacity remained unaltered despite the disrupted assembly of complex V (Fig. 1H and Supplementary Fig. 1G). Additionally, we observed a significant increase in extracellular acidification rate (ECAR), which is typically associated with glycolysis and lactate production (Supplementary Fig. 1H, I). We confirmed the dependence of ECAR on glucose, as indicated by a modest increase in glycolytic rate (Fig. 1I and Supplementary Fig. 1J) after correcting for mitochondrial acidification. When mitochondrial ETC function was disrupted, the resulting increase in glycolysis (a compensatory response to maintain energy production) was similar between groups (Supplementary Fig. 1K), implying that glycolytic capacity is not impaired. Taken together, these results suggest that both basal mitochondrial respiration and glycolysis were similarly upregulated in mutant iPSC-MN.

OxPhos and glycolysis are the primary metabolic pathways for ATP generation and an increase in their activity suggests elevated cellular energy demand or expenditure. This phenomenon parallels the rise in REE observed in patients with mitochondrial OxPhos defects, as well as the increased ATP production rates (cellular equivalent to REE measurements obtained via indirect calorimetry in humans and mice) reported in fibroblasts harboring Complex IV deficiencies[14]. Consistent with this, mutant iPSC-MN exhibited increased total ATP production rates (Fig. 1J and Supplementary Fig. 1L). The total ATP flux, combining contributions from both OxPhos and glycolysis, was increased on average ~35% in mutant iPSC-MN (104.2 pmol ATP/min/DNA) compared to control neurons (76.97 pmol ATP/min/DNA) (Supplementary Fig. 1M). This elevation is very similar to the ~30% increase in resting whole-body oxygen consumption observed in mitochondrial disease patients, as measured by indirect calorimetry and normalized to body weight[14]. The increase in ATP production rates was comparable between glycolysis and OxPhos, indicating no preferential shift toward either pathway (Supplementary Fig. 1N).

The *MT-ATP6* encoded subunit participates in forming the proton pore of the ATP synthase, allowing the re-entry of protons into the mitochondrial matrix. Thus, the increased proton leak (Fig. 1G) could have been a direct consequence of the nonsense variant, causing an imbalance in the proton gradient and disrupting the mitochondrial membrane potential, as previously reported for other pathogenic *MT-ATP6* variants[24,25]. However, mitochondrial membrane potential was not significantly changed in our iPSC-MN (Fig. 1K), which might reflect an adaptation to preserve mitochondrial respiration[26].

In summary, using multiple clones we validated that ~50% mutant heteroplasmy did not alter the motor neuron differentiation ability, but increased their metabolic activity and ATP production rates while maintaining mitochondrial mass and membrane potential. These results suggest hypermetabolism as an adaptive mechanism to meet neuronal energy and/or biosynthetic demands. Thus, this offers the first human neuronal model for studying the consequences of hypermetabolism.

### Hypermetabolism increases neuronal panthotenate consumption

To further explore the metabolic phenotype revealed by the Seahorse XF Analyzer, we investigated central carbon metabolism in iPSC-MN using metabolic tracing experiments with fully labeled glucose for 1, 12, and 48 h (Fig. 2A, B). We observed an increase in the number of significantly changed metabolites over time and focused our analysis on the 48-h labeling as it exhibited the most pronounced differences (Fig. 2C and Supplementary Fig. 2A–J). Aligning with the respirometry experiments, mutant iPSC-MN showed increased accumulation of glycolytic end products, pyruvate and lactate, while using glucose similarly as controls (Fig. 2D). We could also detect an increase in glycolysis-derived amino acids, specifically serine and alanine (Supplementary Fig. 3A). This indicated an increased glycolytic flux. Additionally, several TCA cycle metabolites were significantly elevated in

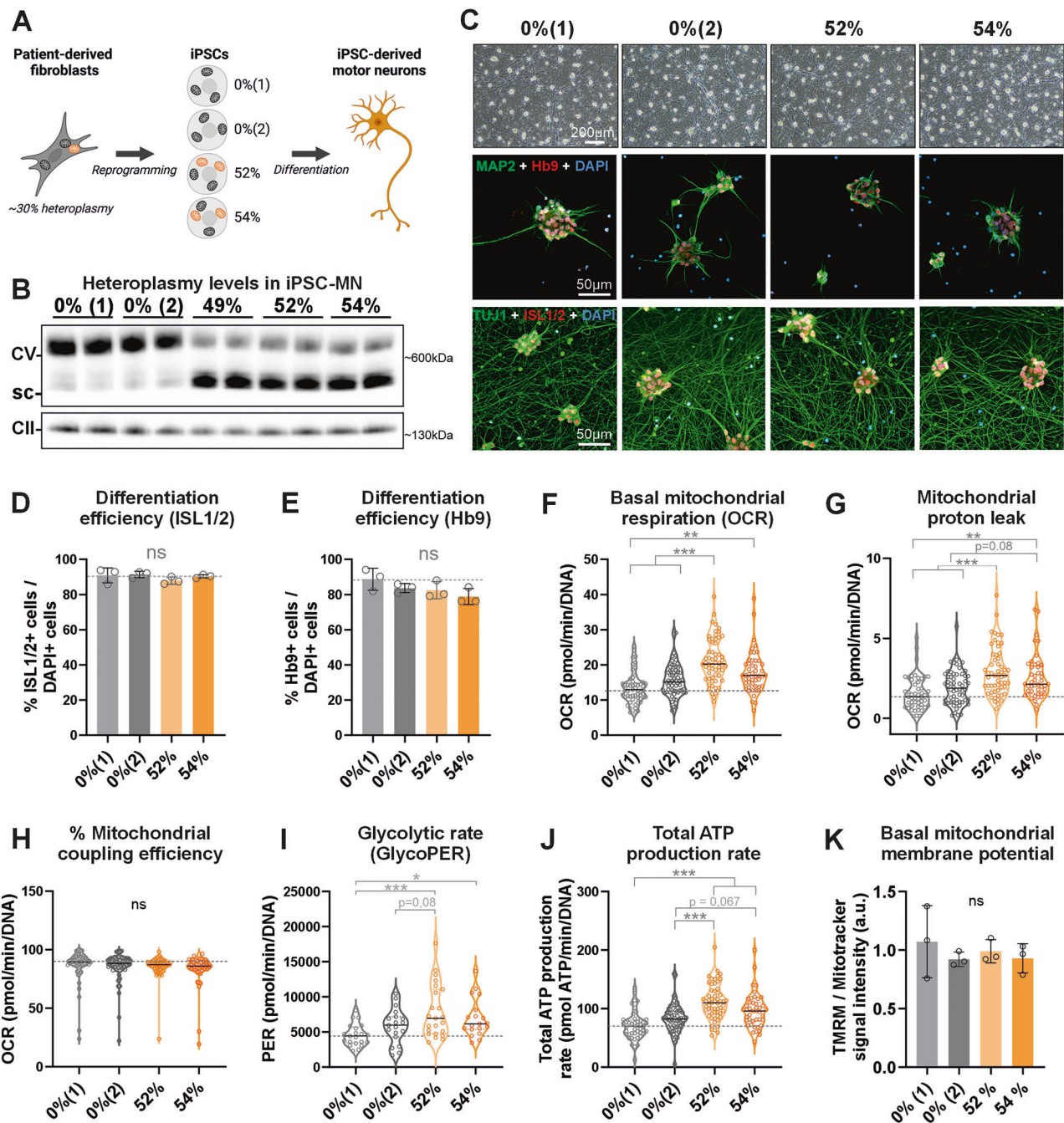

the mutants (Fig. 2E), especially the late part of the cycle (malate and fumarate). The only exception was succinate, which remained unchanged compared to both isogenic controls. The overall enrichment of TCA metabolites suggests an increased demand for mitochondrial ATP synthesis and/or metabolic intermediates for biosynthetic pathways, while the unaltered levels of succinate might reflect increased utilization by respiratory complex II[27].

Most of the significantly changed metabolites were increased in the mutants. However, we noticed that one metabolite, pantothenate, was consistently decreased across all time points (Fig. 2C, F and Supplementary Fig. 2A, B, and Supplementary Fig. 3B, F). Also known as vitamin B5, it is an essential nutrient that cannot be synthesized by cells. It is commonly found in a wide variety of foods and serves as the necessary precursor to coenzyme A (CoA), which is vital for the metabolism of fatty acids, carbohydrates, and amino acids.

Among these pathways, acetyl-CoA synthesis is particularly important (Fig. 2B), as it plays a central role in numerous cellular processes essential for life[28,29]. The increased consumption of pantothenate suggested a higher demand for its downstream metabolic products.

The conversion of pyruvate to lactate is reversible and can also serve as a carbon source for the TCA cycle in neurons[30]. To test whether the increased lactate in mutant neurons resulted from its increased demand as a carbon source, we fed the cells with fully labeled lactate for 48 h. We observed that 20% of the pyruvate was labeled (Supplementary Fig. 3C), confirming that motor neurons can use lactate as fuel. However, no difference in the relative abundance of labeled pyruvate was detected among cell lines (Supplementary Fig. 3D), suggesting that lactate consumption remained unchanged in mutant iPSC-MN. Thus, the higher lactate production might serve for a different purpose. Increased lactate levels are commonly observed in mitochondrial diseases, although their precise

**Fig. 1 | 50% *MT-ATP6* heteroplasmy does not affect motor neuron differentiation but leads to hypermetabolism. A** Schematic cartoon illustrating the experimental design. Patient skin cells with approximately 30% heteroplasmy were previously reprogrammed into iPSC, resulting in clones with varying levels of heteroplasmy[19]. We selected two control lines (0% heteroplasmy) and two lines with ~50% heteroplasmy for differentiation into motor neurons. Created in BioRender. Torregrosa, R. (2025), https://BioRender.com/rijw3bk. **B** Representative cropped immunoblot of Complex V assembly determined by Blue-Native PAGE. Complex V (CV) was detected using anti-ATP5FA1, targeting the F1 subunit (facing the matrix) of ATP synthase; and Complex II (CII) was detected using anti-SDHA, which served as the loading control. $n = 2$ technical replicates (independent wells), sc subcomplexes. **C** iPSC-derived motor neurons on day 30. Upper panels, representative bright field microscope images showing similar neural morphologies, scale bar 200 μm. Lower panels, representative immunocytochemistry images of neuronal markers MAP2 (green), Hb9 (red), TUJ1 (green), ISL1/2 (red) and nucleus (Dapi, blue). Scale bar 50 μm. **D, E** Differentiation efficiency measured as the % of ISL1/2- or HB9-positive nuclei relative to DAPI positive cells from Fig. 1C. Data represent the mean of ~1000 cells quantified from 15 images taken across three independent frames ($n = 3$), obtained from two separate differentiation experiments. **D** % of ISL1/2-positive nuclei relative to DAPI positive cells from Fig. 1C. **E** % Hb9-positive nuclei relative to DAPI positive cells from Fig. 1C. **F** Basal mitochondrial respiration

determined before oligomycin injection (Seahorse XF Mito Stress test). **G** Proton leak was calculated as the remaining mitochondria respiration rate after inhibition of complex I and II by rotenone and antimycin (Seahorse XF Mito Stress test). **H** % Coupling efficiency was determined as the fraction of basal ATP-linked OCR/basal OCR and indicates the proportion of $O_2$ consumed to fuel ATP synthesis (Seahorse XF Mito Stress test). Expressed as %. For (**F–H**) $n = 13$–21 wells per cell line per experiment, from three independent differentiations. Data was normalized to total DNA per well. **I** Glycolytic rate (glycoPER) determined after subtracting mitochondrial acidification from total proton efflux rate (Seahorse XF Glycolytic Rate Assay). $n = 9$–11 wells per cell line per experiment, from two independent experiments. Seahorse data was normalized to total DNA per well. **J** Total ATP production rate (ATP-OxPhos + ATP-glycolysis) computed from the basal OCR and ECAR (Seahorse XF Mito Stress test). **K** Mitochondrial membrane potential based on the intensity of TMRM (membrane potential) normalized to Mitotracker (mitochondrial mass). Each dot represents an independent motor neuron differentiation experiment (n_exp = 3 independent differentiations) with an average of 2 to 11 wells per cell line per experiment. For all graphs, data shown as mean ± standard deviation, *$p < 0.05$, **$p < 0.01$, ***$p < 0.001$, or ns non-significant by One-Way ANOVA followed by Bonferroni´s (Seahorse XF) or Tukey´s (membrane potential) multiple comparison post-hoc test.

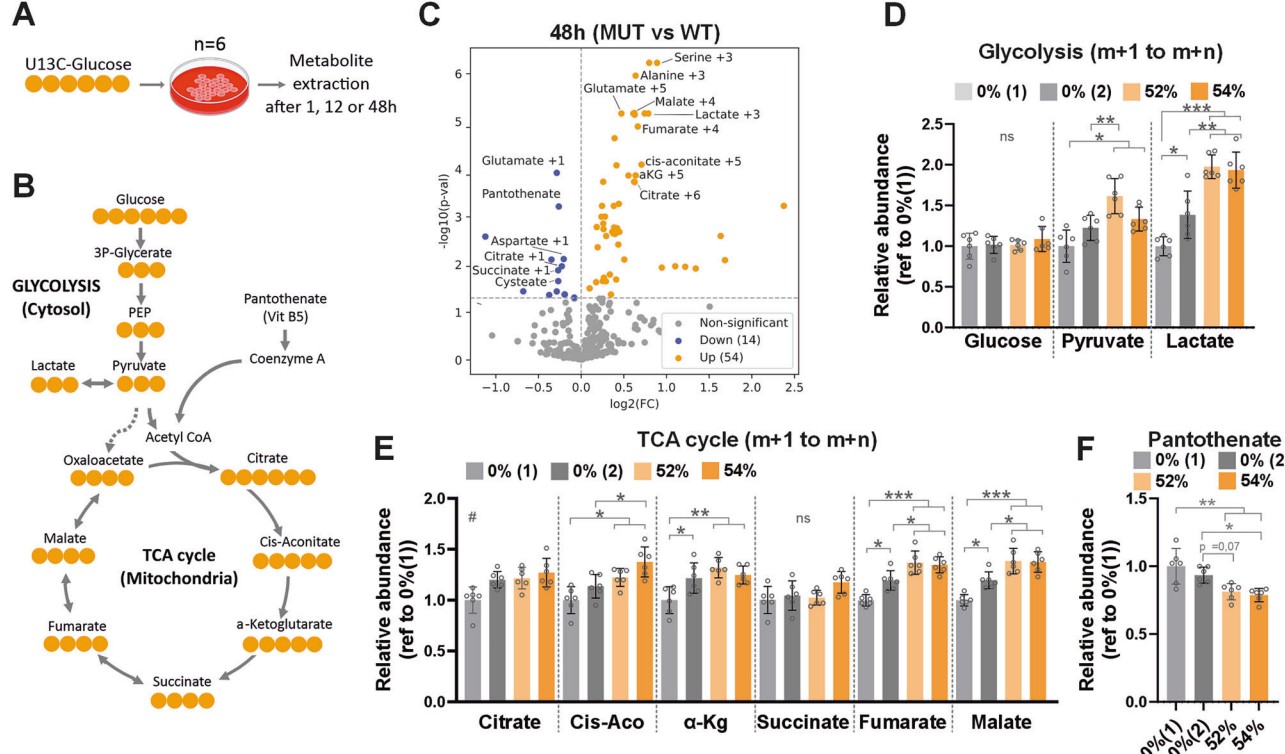

**Fig. 2 | ATP synthase deficiency leads to increased consumption of pantothenate. A** Scheme depicting the metabolic tracing experimental plan. On day 30 of the MN differentiation, the media was replaced with media-containing $U^{13}C$-glucose. Intracellular metabolites were extracted 1, 12 or 48 h later. Six individual wells per cell line were used as replicates. **B** Cartoon showing the expected fate of the labeled carbons derived from $U^{13}C$-Glucose. **C** Volcano plot showing significant changes in metabolites 48 h, comparing pooled mutant samples (52% and 54% mutants, MUT) versus wild-type (samples (0%(1) and 0%(2), WT). Thresholds: FDR = 0.05,

log2(FC) = 1. Only relevant metabolites were labeled. Relative abundance of representative glycolysis (**D**) and OxPhos (**E**) metabolites after 48 h of isotopic labeling, calculated as the total sum of the labeled fraction (m + 1 to m + n) and referred to one control group 0%(1). **F** Relative abundance of pantothenate after 48 h of isotopic labeling. Pantothenate cannot be synthetized within the cell, and therefore is not labeled (m + 0). Data shown as mean ± standard deviation, *$p < 0.05$, **$p < 0.01$, ***$p < 0.001$, # vs all, by One-Way ANOVA followed by Tukey´s multiple comparison post-hoc test.

significance remains unclear. For instance, lactate production might be needed for regenerating $NAD^+$, thereby maintaining the $NAD^+$/NADH ratio and ensuring the continuation of glycolysis (Warburg effect). Alternatively, it has recently proposed that lactate could activate OxPhos through mechanisms that are independent of its metabolic function[31].

In sum, the metabolic tracing analysis supported the hypermetabolic phenotype, revealing an upregulation of the main bioenergetic and

biosynthetic pathways in the cells, glycolysis, and OxPhos. Furthermore, our results indicated a connection between hypermetabolism and increased consumption of pantothenate in neurons, likely reflecting an effort to meet the elevated substrate demands.

We next tested whether avanafil, a PDE5 inhibitor previously shown to rescue ATP synthase deficiency caused by an *MT-ATP6* missense mutation[25], could restore the metabolic changes observed in mutant

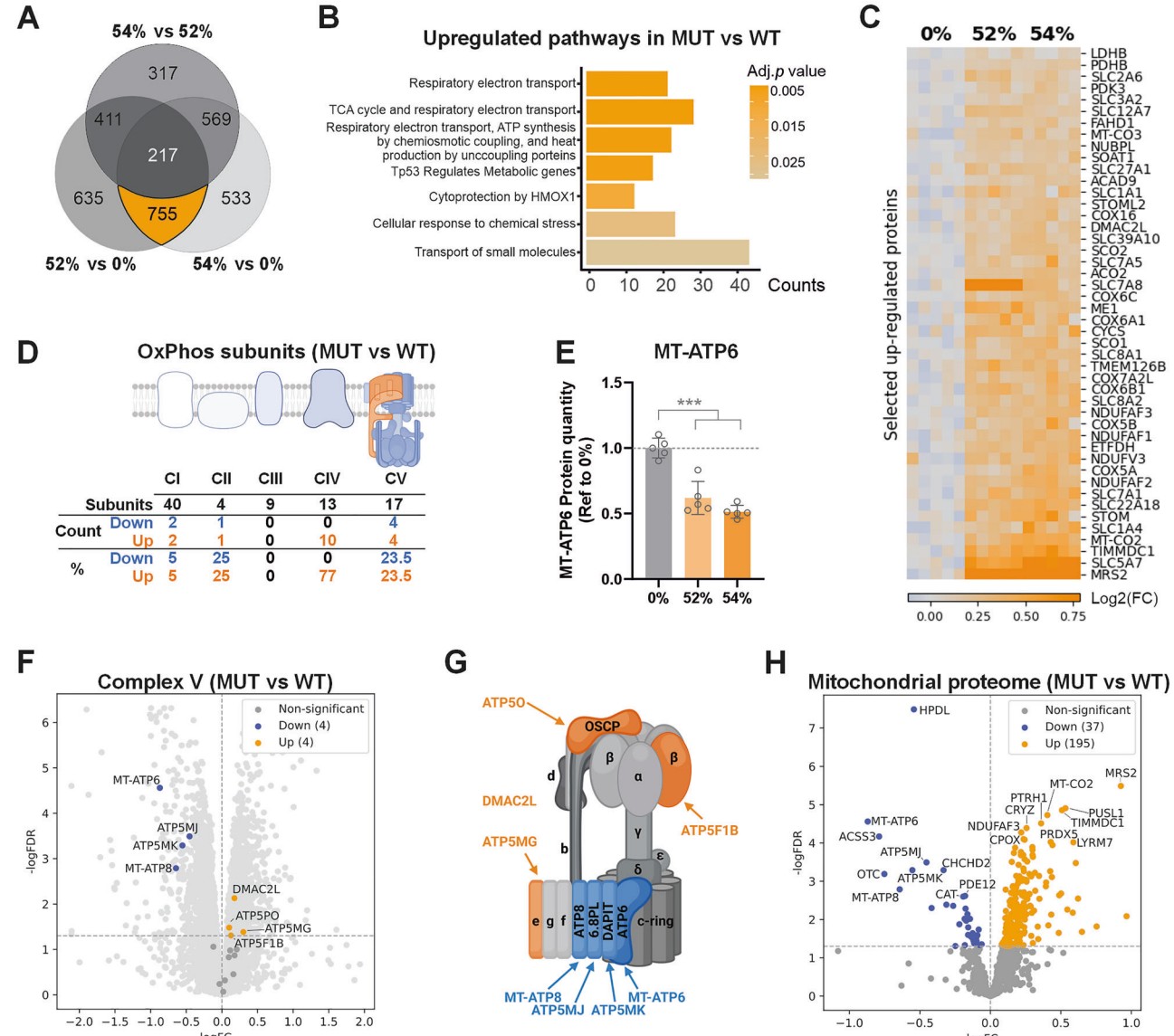

**Fig. 3 | The mitochondrial proteome in neurons is reorganized to support the hypermetabolic state.** **A** Venn diagram reveals a large amount of shared differently expressed protein between mutants and the 0%(1) (755). Proteins with FDR-corrected $p < 0.01$ were considered differentially expressed. **B** Pathway Enrichment Analysis of Reactome terms enriched pathways in MUT (52% + 54%) vs WT (0% (1)). Differentially upregulated pathways are indicated by gene ratio (counts) and FDR (adj.p-value). **C** Heatmap of selected proteins extracted from the enrichment analysis involved in glycolysis, TCA, OxPhos and small molecules transporters. **D** Summary table showing changes in the expression profile of OxPhos subunits for each complex. Blue indicates downregulation, while orange indicates upregulation in mutant MNs. Created in BioRender. Torregrosa, R. (2025) https://BioRender. com/r59eco7. **E** MT-ATP6 protein levels extracted from the proteomic data set. Data shown as mean ± standard deviation, ***$p < 0.001$, by One-Way ANOVA followed by Tukey's multiple comparison post-hoc test. **F** Volcano plot depicting protein expression changes of the ATP synthase subunits between MUT and WT Thresholds: FDR = 0.01, log2(FC) = 1. **G** Cartoon depicting the subunits of ATP synthase based on[20], created with BioRender.com. Blue labels indicate downregulated proteins, while orange labels indicate upregulated ones in mutant iPSC-MN. **H** Volcano plot showing differentially expressed mitochondrial proteins between MUT and WT including the 915 annotated proteins from MitoCarta 3.0 detected in the data set. The top 10 changes are labeled. Thresholds: FDR = 0.01, log2(FC) = 1.

iPSC-MN. However, pantothenate consumption and $^{13}$C-glucose incorporation into lactate and citrate were unchanged in avanafil treated cells (Supplementary Fig. 3E–H), indicating that the drug failed to restore glucose utilization and TCA cycle activity under our experimental conditions.

**Mitochondrial proteome is reorganized to support the hypermetabolic state**

Next, we performed proteomics of one control and the two mutant cell lines to gain a deeper understanding of the mechanisms behind the hypermetabolic adaptation observed in mutant iPSC-MN. We identified a total of 755 shared differentially expressed proteins in mutants versus the

control (Fig. 3A, in orange). According to the enrichment pathway analysis, the top significantly upregulated pathways in mutant iPSC-MN included Reactome terms related to metabolism: respiratory electron transport, TCA cycle, ATP synthesis and transport of small molecules (Fig. 3B). We identified enzymes involved in pyruvate metabolism (LDHB, ME1 and the TCA cycle enzyme ACO2) and fatty acids metabolism (FAHD1, ETFDH and SOAT1) as increased (Fig. 3C). We also detected various small molecule transporters (SLC family) likely needed to maintain the ionic balance in this energetically constrained consumption scenario[14], including amino acid, fatty acid and glucose transporters, as well as the choline transporter (SLC5A7 or ChT) for neurotransmitter synthesis. Additionally, the levels of

multiple OxPhos subunits and assembly factors were elevated in the mutant neurons.

A closer look into the mitochondrial respiratory complexes revealed that complex III was entirely unaffected, while the majority of complex IV detected subunits (77%) were increased in the mutant iPSC-MN (Fig. 3D and Supplementary Fig. 4A). The ATP synthase showed a distinctive pattern with some proteins decreased and others increased. Corresponding to the ~50% heteroplasmy level, the MT-ATP6 protein level was reduced by half in mutants (Fig. 3E). MT-ATP6 is incorporated into the ATP synthase complex, alongside MT-ATP8, when the complex is nearly complete. The incorporation of these two mitochondrial-encoded subunits, MT-ATP6 and MT-ATP8, is stabilized by ATP5MJ (6.8PL). Subsequently, ATP5MK (DAPIT) facilitates the linkage of two ATP synthase monomeric complexes, resulting in the formation of a functional ATP synthase dimer[20,32]. In the mutants, MT-ATP6, MT-ATP8, ATP5MJ, and ATP5MK levels were all decreased (Fig. 3F, G), which may explain the impaired assembly of the ATP synthase (Fig. 1B). In contrast, ATP5PO (OSCP), a protein linking the matrix (F1) and membrane (F0) domains of the ATP synthase, ATP5MG (subunit e), a protein of the F0 domain, and the subunit beta (ATP5F1b)[20] were increased in the mutants. Moreover, DMAC2L, which encodes the regulatory protein Factor B, was also increased. DMAC2L has been suggested to bind the F0 domain of the ATP synthase, where it regulates a second proton translocation pathway, thereby promoting the ATP synthase activity while maintaining the mitochondrial membrane potential[33]. In the neurons, the loss of MT-ATP6 therefore compromised the stability of its assembly partners during the final stages of the ATP synthase complex formation and dimerization, while intriguingly increasing the levels of other subunits. These alterations in the ATP synthase, combined with the increased abundance of complex IV subunits, suggest a restructuring of the OxPhos system.

Following the increases in some OxPhos subunits, we next asked whether this finding represented a global change in the mitochondrial proteome. While there was a bias towards upregulation of mitochondrial proteins (195 proteins, 21.3% of the detected mitochondrial proteins) (Fig. 3H), the increases occurred mostly in the internal mitochondrial subcompartments (matrix and inner membrane) (Supplementary Fig. 4B). In addition, 4% (37 proteins) of the detected mitochondrial proteome showed a significant reduction (Fig. 3H). Rather than indicating an increase in mitochondrial mass, these results suggest a reorganization of the functional mitochondrial proteome. Consistently, we did not see changes in mtDNA copy number (Supplementary Fig. 1B), a proxy of mitochondrial mass.

In summary, our results suggest that the mitochondrial proteome underwent remodeling to adapt to the ATP synthase deficiency likely to support a more active metabolism.

## Hypermetabolism imposes a redistribution of acetyl-CoA utilization

Pantothenate, which was consistently less abundant in *MT-ATP6* mutant motor neurons, is obtained exclusively from the diet via the multivitamin transporter SLC5A6/SMVT42. In our proteomics dataset, the levels of the transporter were not changed (Supplementary Fig. 5A), indicating that pantothenate uptake was not compromised. Since this vitamin is the obligatory precursor of CoA, we investigated the conversion of pantothenate into CoA, but none of the enzymes involved showed any changes (Supplementary Fig. 5A), suggesting that this initial step was also unaffected in mutant iPSC-MN.

CoA acts as a carrier of acyl groups involved in numerous cellular reactions[34]. Of the different fates of CoA (Fig. 4A), the acyl carrier protein (ACP), lysine degradation via α-aminoadipate semialdehyde synthase (AASS), or the folate cycle through 10-formyltetrahydrofolate dehydrogenase (FDH) pathways did not show consistent differences (Supplementary Fig. 5B). However, many of the enzymes involved in the biosynthesis of acetyl-CoA were significantly increased in both mutants (Fig. 4B and Supplementary Fig. 6A)[28,29]. Among those we identified increased levels of proteins involved in mitochondrial pyruvate

decarboxylation (PDHB) and fatty acid catabolism (ACAT1, ACAD, HADH, and ACAA2), as well as cytosolic acetate (ACSS2) and citrate metabolism (ACLY, IDH1, and GOT1). The only exception was the cytosolic ALDH1, involved in aldehyde metabolism, which showed reduced levels. However, its mitochondrial counterpart, ALDH2, was upregulated, suggesting a compensatory balance. Acetyl-CoA levels were slightly elevated but not consistently significant (Fig. 4C), implying an equilibrium between its production and consumption. Therefore, our results suggest that mutant iPSC-MN have an increased demand for pantothenate likely to maintain acetyl-CoA levels, needed to meet cellular requirements.

Next, we focused on the different fates of acetyl-CoA in TCA cycle, lipid synthesis, acetylcholine synthesis and protein acetylation (Fig. 4A). During pyruvate oxidation in mitochondria, the pyruvate dehydrogenase (PDH) complex converts pyruvate into acetyl-CoA, which is then shunted into the TCA cycle, forming citrate. We observed increased levels of one subunit of the PDH complex, PDHB (Fig. 4B). However, the PDH inhibitor PDK3, which phosphorylates and inhibits the catalytic subunit of the PDH complex (PDHA1), was also elevated in the mutants (Fig. 4B). Despite this, we did not find significant differences in the levels of PDHA1 phosphorylation (Supplementary Fig. 6B, C), implying that its activity remained unchanged. In contrast, we observed increased activity of the alternative entry point for pyruvate into the TCA cycle, via pyruvate carboxylase (PC) (Supplementary Fig. 6D–F). This may also explain the greater accumulation of the metabolites in the last part of the TCA cycle, as PC redirects carbons into oxaloacetate, which can then be converted into malate and fumarate (Fig. 2E). These results indicate that mitochondria in mutant iPSC-MN did not oxidize more acetyl-CoA derived from glucose, but instead, the TCA cycle was replenished more extensively through PC.

Then, we explored another fate of acetyl-CoA, lipid synthesis. We observed that the primary cytosolic enzyme involved in fatty acid synthesis, FASN, and the last enzyme of mitochondrial fatty acid synthesis (mtFAS), MECR, were upregulated in mutant motor neurons (Fig. 4D), suggesting that fatty acid biosynthesis using acetyl-CoA was increased in mutant iPSC-MN. While cytosolic fatty acid synthesis is primarily involved in creating lipid pools for energy storage and membrane formation, mtFAS is believed to play a more regulatory role[35,36]. One of the functions of mtFAS is the production of lipoic acid, which is critical in modulating the activity of several dehydrogenase enzymes, including the PDH complex and oxoglutarate dehydrogenase (OGDH)[37]. However, we did not observe any change in the levels of lipoylated proteins (Supplementary Fig. 6G, H), consistent with our other findings suggesting that mutant iPSC-MN did not oxidize more acetyl-CoA in the mitochondrial TCA cycle. On the other hand, we observed increased levels of ACLY and ACSS2 (Fig. 4B and Supplementary Fig. 6A), which convert acetyl-CoA in the cytosol and nucleus, supporting an increased demand for this metabolite in these compartments.

In motor neurons an important fate of cytosolic acetyl-CoA is acetylcholine (Ach), the primary neurotransmitter[38], which is synthesized in nerve terminals from acetyl-CoA and choline. If Ach synthesis was compromised, its low levels could potentially contribute to the patient's neuropathy phenotype. However, the levels of the neurotransmitter were not reduced in mutant iPSC-MN (Fig. 4E and Supplementary Fig. 7A). Yet we found that several enzymes involved in the processing of ACh were increased, particularly the transporters VAChT and ChT and its recycling enzyme AChE (Fig. 4F), suggesting a need for compensatory upregulation to maintain the neurotransmitter levels and a potential vulnerability for neuronal function.

Cytosolic acetyl-CoA also serves as a crucial cofactor in post-translational modifications of proteins, essential for lysine acetylation of histones and other proteins. Histone acetylation by histone acetyltransferases (HATs) represents a fundamental mechanism of epigenetic regulation in neurogenesis and neurodegeneration[39]. In this regard, among the top downregulated pathways identified in our proteomics dataset, we found Reactome terms related to gene expression, including chromatin organization, epigenetic regulation, acetylation of histones by HATs and DNA methylation (Fig. 4G). Downregulated proteins in the mutant

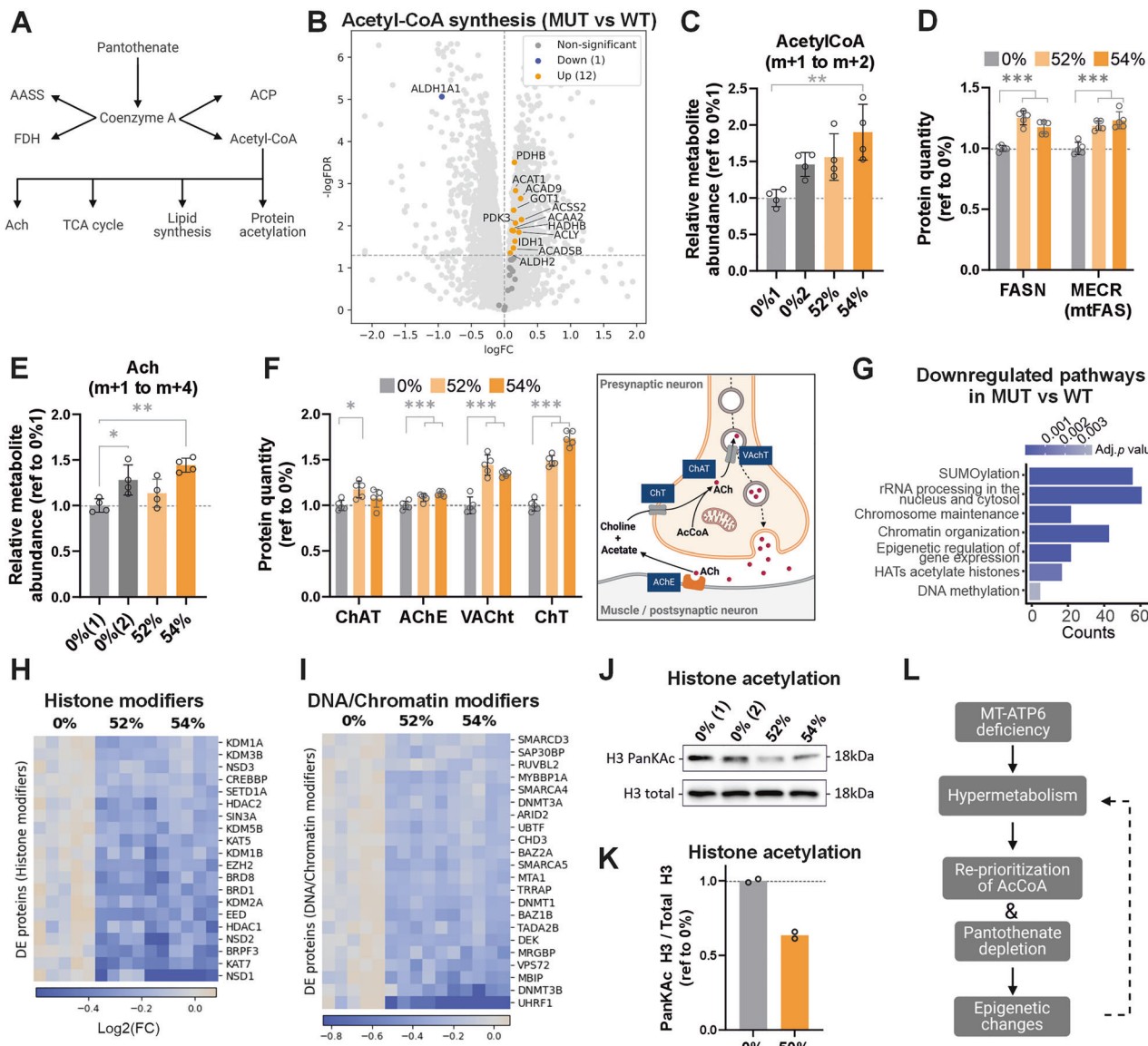

**Fig. 4 | Hypermetabolism leads to the reprioritization of acetyl-CoA in neurons.**
**A** Fates of coenzyme A, the primary product of pantothenate, including lipid bio-synthesis through acyl carrier protein (ACP), lysine degradation via α-aminoadipate semialdehyde synthase (AASS), the folate cycle through 10-formyltetrahydrofolate dehydrogenase (FDH), and acetyl-CoA biosynthesis. Acetyl-CoA serves as a metabolic hub involved in acetylcholine (ACh) production, bioenergetics (TCA cycle), lipid biosynthesis and protein acetylation. **B** Volcano plot showing protein expression changes of proteins involved in acetyl-CoA biosynthesis between MUT and WT. Thresholds: FDR = 0.01, log2(FC) = 1. **C** Relative abundance of acetyl-CoA measured using UC$^{13}$-glucose metabolic tracing, calculated as the total sum of the labeled fraction (m + 1 and m + 2). **D** Expression levels of FASN, involved in fatty acid biosynthesis in the cytosol and MECR, involved in the last step of fatty acid biosynthesis in mitochondria. **E** Relative abundance of acetylcholine measured using UC$^{13}$-glucose metabolic tracing, calculated as the total sum of the labeled fraction (m + 1 to m + 4). **F** Left, expression levels of proteins involved in acetylcholine synthesis and processing. Referred to the control group. Right, scheme depicting the synthesis, transport and hydrolysis of the neurotransmitter acetylcholine (Ach)[38].

Created in BioRender. Torregrosa, R. (2025), https://BioRender.com/0gy0ypl.
**G** Pathway Enrichment Analysis of Reactome terms enriched pathways in MUT (52 + 54%) vs WT. Differentially downregulated pathways are indicated by gene ratio (counts) and FDR (adj.p-value). **H** Heatmap of selected proteins related to histone modifications. **I** Heatmap of selected proteins related to DNA modifiers and chromatin remodeling. Data shown as mean ± standard deviation, *$p < 0.05$, **$p < 0.01$, ***$p < 0.001$, by One-Way ANOVA followed by Tukey´s multiple comparison post-hoc test. **J** Representative cropped immunoblot of total H3 and pan-acetylated (PanKAc) H3 in iPSC-derived MN. **K** Quantification of (**J**) as the ratio total H3/ PanKAc H3. $n = 2$ biological replicates (cell lines). **L** Proposed model: ATP synthase deficiency caused by the truncating *MT-ATP6* mutation triggers a metabolic rewiring, leading to hypermetabolism. This reflects an increased con-sumption of pantothenate possibly due to increased demands of acetyl-CoA for biosynthetic pathways, limiting the availability for histone acetylation. Conse-quently, this drives epigenetic changes influencing gene expression and likely modulating an adaptation.

iPSC-MN included histone modifiers involved in the addition or removal of acetyl groups (BRD1, BRPF3, CREBBP, EPC1, KAT7, HDAC1, and HDAC2) and methyl groups (EZH2, KTM2A, and several KDM family proteins) (Fig. 4H). Additionally, DNA methylases (DNMT3A, DNMT3B, and DNMT1), as well as chromatin remodelers, transcriptional regulators, and coactivators (UHRF1, CHD3, TDG, VPS72, DEK, and several SMARC

family proteins) were downregulated (Fig. 4I). The downregulation of protein acetylation seemed specific to lysine acetylation, as the N-terminal acetylases remained largely unaffected (Supplementary Fig. 8A). We also observed reduced levels of two key histone deacetylases, HDAC1 and HDAC2, which may be required to preserve the acetylated histones that remain. To determine whether histone acetylation was reduced in mutant

neurons, we measured the pan-acetylation levels of histone H3, confirming a ~ 40% reduction in the mutants (Fig. 4J, K and Supplementary Fig. 8B) and highlighting a significant change in epigenetic histone acetylation in association with the hypermetabolic phenotype.

The pathway enrichment analysis also revealed that SUMOylation was among the primary downregulated pathways in mutant motor neurons (Fig. 4G). SUMOylation is a post-translational modification that has been linked to various cellular processes, including mitochondrial stress responses[40], metabolic regulation[41–43] and primarily gene expression[44]. The downregulation of SUMOylation could have significant implications for shaping or responding to hypermetabolism, presenting an area that warrants further investigation.

In summary, we propose that the reduced pantothenate levels observed in mutant iPSC-MN, driven by hypermetabolism, reflect an increased demand for and redistribution of acetyl-CoA utilization. If acetyl-CoA is predominantly directed toward supporting biosynthetic pathways, including lipid synthesis, its availability for histone acetylation may become limited, potentially triggering a recalibration of the epigenetic landscape (Fig. 4L). This shift could alter chromatin accessibility and gene expression patterns, reflecting the cell's need to balance metabolic demands with epigenetic regulation under changing conditions. Consequently, the reduction in histone acetylation, and the potential vulnerability in maintaining ACh levels, may reflect trade-offs resulting from the reprioritization of acetyl-CoA.

## Discussion

Hypermetabolism involves molecular adaptations required to sustain cellular functionality due to abnormal and futile energy expenditure[13]. Thus, disease manifestations may arise from increased molecular entropy due to excessive energy production and expenditure and/or energy trade-offs resulting from the reprioritization of energy and metabolism. Our study, pioneer in using a metabolic approach to investigate the consequences of hypermetabolism, and the first in human neurons, supports these two scenarios and highlights acetyl-CoA as a central player. We observed a general increase in acetyl-CoA biosynthetic pathways, while the only notable downregulated metabolite was pantothenate, its precursor. The increased demand for acetyl-CoA likely arose from the abnormal increased metabolism observed in mutant iPSC-MN, leading to changes in the prioritization of its usage and to cellular trade-offs. Hypermetabolism may pose a risk to cells with a high energy demand and to individuals with mitochondrial deficiency who already struggle to maintain adequate levels of energy. As a result, targeting and reducing hypermetabolism could be a potential therapeutic approach.

Pantothenate, a common nutrient found in most food types and an obligatory precursor of CoA, is important for neuronal health. Defects in disease genes involved in CoA synthesis from pantothenate underlie two subtypes of neurodegeneration with brain iron accumulation (NBIA) disease[34,45]. Neurodegeneration resulting from pantothenate deficiency, primarily due to reduced CoA levels[46–49], typically compromises mitochondrial metabolism and impairs the production of ACh, the key neurotransmitter in motor neurons. Although we did not detect a decline in ACh levels in the mutant iPSC-MN, the most part of its processing machinery was significantly upregulated. This suggests an adaptive mechanism to maintain ACh levels, subjecting neurons vulnerable to conditions where acetyl-CoA is limited and prioritized for other pathways[50], which could contribute to the dysfunction of peripheral nerves as well as cholinergic neurons of the central nervous system. Due to its abundance in the diet, pantothenate levels are rarely measured in routine diagnostics. However, based on our results it would be interesting to investigate whether reduction in pantothenate could serve as a biomarker for hypermetabolism in patients. In our iPSC-MN culture model, acetyl-CoA levels were maintained, indicating that there was no shortage of either pantothenate or acetyl-CoA. Rather, we observed differences in the way these metabolites were used. Therefore, supplementation with pantothenate might not be enough to alleviate the cellular changes associated with hypermetabolism.

The first study to reveal hypermetabolism due to impaired OxPhos function described a state of mitochondrial uncoupling[10]. Similarly, another study of hypermetabolism on an ATP synthase variant (ATP5F1B) in patient-derived fibroblasts and cancer cells identified an uncoupling phenotype characterized by increased mitochondrial respiration and proton leak[11]. However, a recent study described models of hypermetabolism resulting from OxPhos dysfunction, which were not attributed to mitochondrial uncoupling[14]. Collectively, these findings suggest that hypermetabolism can be present in conditions that are independent of uncoupling, which is supported by our study. In our neuronal model, the MT-ATP6 nonsense variant resulted in elevated proton leak, likely caused by the altered ATP synthase assembly. However, coupling efficiency as well as mitochondrial membrane potential were unaffected. Thus, in line with previous work[13,14], our results suggest that alterations in energy consumption and reprioritization of resources are the primary drivers of the hypermetabolic phenotype.

Our results suggest that certain ATP synthase defects may particularly predispose to hypermetabolism, although it is known that some MT-ATP6 missense variants clearly lead to hypometabolism[11,15,25]. The mechanisms behind these differential consequences require additional investigations. Of note, although the activation of the integrated stress response and the upregulation of transcriptional or translational stress pathways have been proposed as potential contributors to hypermetabolism[14], we did not detect these in our neuronal model. The stress responses might depend on the nature of the driver mutation, the cell type and/or stage of the phenotype but may not to be relevant in neurons.

Our findings indicate that iPSC-MN with ~50% heteroplasmy undergo significant proteomic changes to sustain a hypermetabolic state. Determining the causal direction of these changes, however, is challenging. While the proteomic remodeling may drive hypermetabolism, it is also possible that the hypermetabolic state imposes new demands that trigger additional proteomic adaptations. This scenario would suggest a reciprocal relationship, where both processes influence each other in a continuous, dynamic feedback loop rather than following a linear cause-and-effect model.

Metabolism is known to modulate epigenetics in response to fluctuations in metabolic activities by providing cofactors and substrates necessary for modifying DNA and histones[51]. Consequently, defects in OxPhos, including hypermetabolism, are often associated with DNA hypomethylation[14,52]. We observed the downregulation of several key factors involved in DNA methylation, supporting the association between hypomethylation and hypermetabolism also in neurons. Moreover, our neuronal model expands the understanding of epigenetic remodeling associated with hypermetabolism by implying changes in histone modifications. The availability of acetyl-CoA, a critical substrate for histone acetylation, can influence epigenetic regulation[53]. We observed the downregulation of several histone acetylases responsible for transferring acetyl groups from acetyl-CoA to histones and reduced levels of histone H3 pan-acetylation suggesting reduced histone acetylation as a possible trade-off of acetyl-CoA reprioritization. Our results also suggest that one of the primary fates of acetyl-CoA was fatty acid synthesis, including both cytosolic and mitochondrial pathways. We did not observe changes in mitochondrial protein lipoylation, but the mitochondrial ACP-acyl long-chain synthesis, which is crucial for stabilizing the ETC, may be another pathway demanding acetyl-CoA[35].

In conclusion, using patient-derived iPSC-MN we demonstrate that a nonsense variant in the MT-ATP6 gene, affecting the assembly of ATP synthase, leads to hypermetabolism characterized by increased mitochondrial activity but unaffected coupling between the ETC and ATP synthesis. Our findings suggest that this metabolic rewiring leads to the reprioritization of the central metabolite acetyl-CoA, with histone acetylation representing a trade-off and posing ACh synthesis as a vulnerability for motor neurons. These results contribute to the emerging concept of hypermetabolism in mitochondrial diseases, describing it in neurons for the first time to the best of our knowledge, and highlighting pantothenate depletion and the preference for acetyl-CoA use as signatures. Moreover, this work paves the way for new research avenues to better understand the interplay between metabolism and neurodegeneration.

## Methods

### Experimental model

**Cell lines.** Human iPSC reprogrammed from skin fibroblasts of a patient with peripheral neuropathy and cerebellar ataxia carrying a heteroplasmic *MT-ATP6* nonsense variant m.9154 C > T (p.Gln210Ter) were reported previously[19]. Two isogenic controls with 0% heteroplasmy (0% (1) and 0%(2)) and up to three mutants with 49%, 52% or 54% heteroplasmy were used in this study. All the iPSC clones had comparable morphology and expressed similarly the pluripotency markers. Human patient stem cell research was approved by the Coordinating Ethics Committee of the Helsinki and Uusimaa Hospital District (Nr 95/13/03/00/15).

**Stem cell culture.** iPSC were cultured at 37 °C, in a humidified atmosphere, normoxia and 5% $CO_2$; and maintained on Matrigel-coated plates (Corning, #CLS354234) with E8-medium (Gibco, #A1517001) supplemented with E8-supplement, 50 µg/mL Uridine (Sigma, #U3003) and 100 µg/mL Primocin™ (InvivoGen, #ant-pm-05). iPSC were passaged once or twice a week with 0.5 mM EDTA in PBS when 60% confluent. Experiments were done with cells from passage numbers between 14 and 30.

### iPSC-motor neuron differentiation

iPSC were differentiated into motor neurons following a protocol based on[54,55], with some modifications (Supplementary Fig. 1A). Neuronal basal medium was prepared as 50% DMEM/F-12 (Gibco #31331-028) and 50% Neurobasal® (Gibco #21103-049), and supplemented with N2 (Gibco #17502048), B-27 (Gibco # 17504044), 0.1mM L-ascorbic acid (Santa Cruz #sc-394304), 50 µg/mL Uridine and 100 µg/mL primocin. For the initial neural induction, iPSCs were dissociated using EDTA and plated on ultra-low attachment dishes (Nunclon Sphera 6-well Plate, 174932 Thermo-Fisher). On day 0, neuronal basal medium was supplemented with 3 µM Chir-99021 (Selleckchem # S1263), 40 µM SB431542 (Millipore # 616461), 0.2 µM LDN-193189 (Sigma # SML0559) and 5 µM Y-27632 (Selleckhem # S1049). The medium was replaced the following day. From day 2 to 6, media was changed to neuronal basal medium supplemented with 0.1 µM retinoic acid (ThermoFisher #044540.04) and 0.5 µM SAG (Millipore # 566660). On day 7, medium was replaced with neuronal basal medium supplemented with retinoic acid and SAG as before, 10 ng/ml BDNF (Peprotech #450-02) and GDNF 10 ng/ml (Peprotech # 450-10). On day 9, medium was replaced as previously, plus 20 µM DAPT (Calbiochem #565770). On day 10, the motor neuron spheres were dissociated into single cells with Accumax (Invitrogen 00-4666-56), and motor neuron progenitors were plated on 50 µg/ml poly-D-lysine (Merck Millipore #A-003-E) and laminin 10 µg/ml (Sigma-Aldrich #L2020) coated plates at 90,000 cells/cm². On day 11, fresh medium was added to the wells. On day 14, half of the medium was replaced with neuronal basal medium supplemented only with DAPT, BDNF, and GDNF. From day 16 on, half of the culture medium was replaced with neuronal basal medium supplemented with 10 ng/ml BDNF, GDNF and CNTF (Peprotech 450-13). All the experiments were performed on day 30 mature motor neurons, unless otherwise indicated. Occasionally, "flat cells" appear in the motor neuron culture, however, we carefully monitored that the wells used in this study did not containing flat cells to avoid confounding results. For the avanafil treatments, 1 µM Avanafil (Selleckchem #S4019) was added to the media during the indicated times.

### Method details

**Immunocytochemistry.** Cells were cultured on cover glasses and fixed with 4% paraformaldehyde (PFA) for 15 min at RT and then permeabilized with PBS containing 0.2% Triton X-100 (FisherScientific #BP151-100) for 10 min. Cells were blocked with 5% protease-free BSA (Biowest # P6154) in 0.1% Tween20 (Sigma, #P1379) in PBS-T for 2 h in RT, and then were incubated overnight at +4 °C in blocking buffer containing 5% bovine serum and primary antibodies: HB9 (DSHB #81.5C10-s, 1:50), ISL1 (DSHB #39.4D5-s, 1:50), MAP2 (Abcam #5392, 1:1000), TUJ1 (Biolegend #801201, 1:1000), total H3 (Cell Signaling #4499, 1:200) and PanKAc H3 (Abcam #ab47915, 1:400). Next day, cells were washed with PBS-T and then incubated with the secondary antibody diluted 1:1000 for 1 h at RT: AlexaFluor™ 594 anti-rabbit IgG (A11012, Thermo Fisher), AlexaFluor™ 488 anti-mouse IgG (A11008, Thermo Fisher), AlexaFluor™ 488 anti-goat IgG (A11055, Thermo Fisher), DyLight 488 anti-chicken IgG (SA5-10070, Thermo Fisher). Cover glasses were applied on microscope slides with Vectashield DAPI (VectorLabs #H-1200) and imaged with Axio Observer Z1 (Zeiss) inverted fluorescence microscope. Validation of MN differentiation efficiency on day 30 was calculated as the number of ISL1/2 or Hb9 positive cells per nuclei (DAPI positive) using an automatized and unbiased approach with ImageJ[56]. Data represent the mean of ~1000 cells quantified from 15 images taken across three independent frames ($n = 3$), obtained from two separate differentiation experiments.

**Restriction fragment length polymorphism (RFLP).** Fluorescent RFLP (fRFLP) was previously used to accurately determine mtDNA heteroplasmy levels in iPSC clones[19]. Here, standard RFLP was used to validate heteroplasmy levels in motor neurons. Total cellular DNA was extracted using NucleoSpin DNA extraction kit (Macherey-Nagel #740952.50) and then 60 ng of DNA was amplified by PCR using MyTaq HS Red Mix (Bioline #BIO-25047) and 0.375 µM per primer (fw: CCATGGC-CATCCCCTTATGA and rv: GGTCATGGGCTGGGTTTTAC). The following thermo profile was used: 1 min at 95 °C followed by 40 cycles of 3-step amplification: 15 s at 95 °C, 15 s at 63 °C and 10 s at 72 °C. Next, samples were digested with FastDigestHpyF3I restriction enzyme (ThermoFisher). Half of each sample was separated on standard 2% agarose gel with SYBR™safe DNA gel stain (Invitrogen # S33102) at 110 V for 60 min. The intensity of the bands was captured with Chemidoc XRS + (Bio-Rad) and Image Lab Software (BioRad), and approximated levels of heteroplasmy were determined as the fraction of mutant peak intensity/wild type (wt) + mutant (mut).

**Western blot.** Total protein was obtained from motor neurons by lysing cells in RIPA buffer (ThermoFisher # 89900) supplemented with 1x Halt™ protease inhibitor (Thermo Scientific #87786). Protein concentration was quantified using the BCA Protein Assay Kit (Pierce, #23225) according to the manufacturer's instructions. Then, 10 µg of protein per sample were prepared in 4x Laemmli buffer (BioRad # 1610747), boiled for 5 min at 95 °C and separated on 4–20% Mini-PROTEAN® TGX™ Precast Protein Gels (BioRad #4561094). Proteins were transferred to a Trans-Blot Turbo Mini 0.2 µm Nitrocellulose (BioRad #1704158) using the Trans-Blot Turbo transfer system (Bio-Rad). Membranes were blocked with 5% milk (or 3% BSA for phosphorylated proteins) in TBS-T for 1 h at room temperature, followed by overnight incubation with primary antibodies (SDHA (Abcam #ab14715, 1:2000), ATP5A (Abcam #ab14748, 1:1000), PDHA1 (Abcam #ab110330, 1:2000), S293 PDHA1 (Abcam #ab177461, 1:2000), TUJ1 (Biolegend #801201, 1:1000), lipoic acid (Calbiochem #437695, 1:2000), total H3 (Cell Signaling #4499, 1:200) and PanKAc H3 (Abcam #ab47915, 1:400), all prepared in 3% BSA in TBS-T). The next day, after three washes of 20 min each with TBS-T, membranes were incubated with secondary antibodies for 1 h (Peroxidase Goat Anti-Mouse IgG (Jackson #115-035-146, 1:5000), Peroxidase Goat Anti-rabbit IgG (Jackson #111-035-003, 1:5000), all prepared in 3% BSA in TBS-T). Following another three 20-min washes with TBS-T, the luminescent signal was detected using ECL reagent (Thermo Scientific), and membranes were imaged with the ChemiDoc XRS+ (Bio-Rad). Band intensities were quantified using Image Lab Software (Bio-Rad). For the analysis, total protein was determined by Ponceau staining (Fisher Scientific #33427.01) or, alternatively, TUJ1 signal was used as a loading control.

**Blue-Native PAGE.** The assembly of the mitochondrial respiratory complex V in iPSC and iPSC-MN was studied with Blue-Native-PAGE

using isolated mitochondria. To obtain enough material, at least two 6 cm dishes were combined to create a single sample. After collection, cells were sonicated, and protein concentration was determined using the Bradford method. Then, 5 mg/mL protein samples were treated with the same volume of 4 mg/mL digitonin (Sigma #D141) for five minutes. The reaction was stopped by adding PBS up to 1.5 mL, and samples were pelleted for 10 min at 10,000 g. Pellets were resuspended in Blue-Native gel buffer (1.5 M aminocaproic acid, 150 mM Bis-Tris, pH 7.0), supplemented with 1 M aminocaproic acid and 2 mM EDTA. Then, 10% lauryl maltoside (0.1 g/mL) in PBS was added and samples were incubated for 2 h on ice. After this, samples were pelleted for 20 min at 20,000 g in cold, and the supernatant was transferred into a new Eppendorf tube. Samples were used immediately or frozen at −80 °C. Separation of 10 μg of proteins by Blue-Native-PAGE was performed as described before[57].

**mtDNA copy number.** Total cellular DNA was isolated using proteinase K (ThermoFisher # E00491) and SDS lysis (10 mM Tris pH 7.4, 10 mM EDTA, 150 mM NaCl and 0.4% SDS) followed by phenol:chloroform extraction and ethanol precipitation[58] using Ultrapure Phenol:-Chloroform:Isoamyl Alcohol (ThermoFisher #15593031). To increase the efficiency of the PCR reaction and prevent variations caused by different DNA topologies, total DNA samples were gently sonicated using an ultrasonic processor (VibraCell, TAMRO) set at 50% duty cycle. Total DNA was measured using Qubit™ dsDNA Quantification Assay Kit (Invitrogen, #Q32850) and QFX Fluorometer (DeNovix). Levels of mitochondrial (*MT-ND5*) and nuclear (*B2M*) DNA were analyzed by qPCR amplification in CFX Real-time system C1000Touch (Bio-Rad) with DyNAmo Flash SYBR Green qPCR Kit (ThermoFisher # F415L), using 0.5 μM per primer and 2 ng of DNA for nuclear DNA or 0.2 ng of DNA for mtDNA reactions[59]. The following thermoprofile was used: 7 min at 95 °C followed by 40 cycles of a 2-step amplification: denaturation at 95 °C for 10 s and annealing and elongation at 60 °C for 30 s. Two technical replicates per sample were measured. The primers used were *B2M* (Forward: TGCTGTCTCCATGTTTGATGTATCT, Reverse: TCTCTGCTCCCCACCTCTAAGT) and *MT-ND5* (Forward: AGGCG CTATCACCACTCTGTTCG, Reverse: AACCTGTGAGGAAAGGTAT TCCTG).

**Seahorse XF analysis.** Mitochondrial oxygen consumption (OCR), extracellular acidification rate (ECAR) and proton efflux rate (PER) in iPSC-MN were determined using Seahorse XF96 Extracellular Flux Analyzer (Agilent), employing the Agilent Mito Stress Test and Glycolytic Rate Assay. Briefly, on day 10 of the MN differentiation, 20,000 cells were seeded on poly-D-lysine and laminin coated XF96 Seahorse plates, and cells were maintained up to day 30 following the differentiation protocol as usual. For the Mito Stress Test, at day 30, OCR was measured using 2 μM oligomycin (Sigma # O4876), 5 μM FCCP (Sigma #C2920) and 1 μM rotenone (Sigma #R8875) and antimycin-A (Sigma #A8674), employing Seahorse basal media supplemented with 25 mM glucose, 1 mM pyruvate and 2 mM glutamine. For the Glycolytic Rate Assay, at day 30, ECAR was measured using 1 μM rotenone and antimycin-A, and 50 mM 2-Deoxyglucose (Sigma #D8375), employing the same basal media as before. For the Mito Stress Test, three independent differentiation experiments were performed, including 13–21 wells per cell line per experiment. For the Glycolytic Rate Assay, two independent experiments, including 9–11 wells per cell line, were performed. Only wells with a homogeneous monolayer of iPSC-MN were included in the analysis. Data was normalized to DNA concentration using CyQUANT kit (Invitrogen, # C35007). ATP Production Rate was calculated from the Mito Stress Test data after calculating the buffering power of our assay media and following[60].

**Mitochondrial membrane potential (MMP).** On day 30, the differentiation medium was replaced with fresh culture medium containing 50 nM TMRM (Invitrogen #T668) for MMP measurement and 50 nM

Mitotracker Green (Invitrogen #M7514) for mitochondrial mass assessment. Cells were incubated for 30 min. After incubation, the staining medium was removed, the cells were washed once with PBS, and HBSS (Gibco #14025-092) was added for imaging. Red and green signals were measured using the Opera Phenix Plus High-Content Screening System (PerkinElmerRevvity). Images were acquired in confocal mode using a long working distance 10x NA 0.3 air objective and two Andor Zyla sCMOS cameras (2160 × 2160 pixels, 6.5 μm pixel size). The excitation time was set to 300 ms for the 488 nm laser and 400 ms for the 561 nm red laser. Analyses were performed with Harmony 4.9 software using following standard pipelines. For each motor neuron differentiation (n_exp = 3 independent differentiations), the average value across all wells for each cell line was calculated (*n* = between 2 and 11 wells), resulting in a single data point per differentiation.

**Acetylcholine levels (fluorometric kit).** Total acetylcholine levels in day 36 motor neurons were measured using a fluorometric kit (Abcam, #Ab65345) by following the manufacturer's instructions. To ensure an adequate number of cells for each measurement, two 3.5 cm dishes, each containing approximately 1.5 million cells, were combined to form a single sample. The experiment included three independent samples, and each sample was measured in duplicate. Acetylcholine levels were normalized to protein concentration determined using the BCA Protein Assay Kit (Pierce, #23225).

**Histone purification.** Histones were purified using an acid extraction method[61] with slight modifications. To ensure an adequate number of cells for the purification, three 3.5 cm dishes, each containing approximately 1.5 million cells, were combined to form a single sample. Briefly, cells were disrupted with histone lysis buffer (20 mM HEPES buffer pH 7.4, 0.25 M sucrose, 1.5 mM MgCl₂, 2 mM KCl, 0.5% Triton X-100, and 1X HALT™ proteinase and phosphatase inhibitor cocktail), followed by centrifugation at 2000 × *g* and 4 °C for 10 min. The pellet was resuspended in the same buffer and centrifuged under the same conditions. The resulting pellets were suspended in 0.2 M sulfuric acid and left on ice for 2 h with occasional agitation. The suspension was then spun down at 12,000 × *g* and 4 °C for 10 mi. The supernatant was mixed with a 10-fold volume of ice-cold ethanol and left at −20 °C overnight. After centrifugation at 12,000 × *g* and 4 °C for 10 min, pellets (histone-enriched fraction) were resuspended in ice-cold acetone and centrifuged under the same conditions. After three times washing with ice-cold acetone, the histones were dried for 5 min and then dissolved in histone elution buffer (0.0625 M Tris-HCl buffer pH 6.8, 2% SDS, and 10% glycerol). Concentration was calculated using the BCA Protein Assay Kit (Pierce, #23225) and 8 μg of purified histones were used for Western blotting as described before.

**High-throughput methods**
**Metabolic profiling.** For metabolic profiling experiments, iPSC-MN were incubated with 21.25 mM U-¹³C glucose (Cambridge Isotope Laboratories # CLM-1396) and 2 mM glutamine in 50% DMEM-F12 (w/o glucose, Biowest, #L0091) plus 50% Neurobasal®-A (w/o glucose, Gibco, #A24775-01) for 1, 12 or 48 h starting from day 30. Concentration of glucose and glutamine were comparable to normal culture conditions. Then, metabolites were extracted: cells were washed once with PBS and 150 μl of ice-cold H₂O/acetonitrile (Sigma # 34851) (10:40) was added. Cells were scraped, collected, and vortexed for 5 s. Finally, extracted samples were centrifuged at 13,000 *g* for 10 min at 4 °C, and the supernatant was stored at −80 °C until analysis. For each cell line, 6 separate wells were used as technical replicates.

Extracted metabolites were analyzed with a Thermo Q Exactive Focus Quadrupole Orbitrap mass spectrometer coupled with a Thermo Dionex UltiMate 3000 HPLC system (ThermoFisher Scientific). The HPLC was equipped with a hydrophilic ZIC-pHILIC column (150 × 2.1 mm, 5 μm) with a ZIC-pHILIC guard column (20 × 2.1 mm, 5 μm, Merck Sequant). A

5 μl sample was injected into the LC–MS instrument after quality controls in randomized order having every tenth samples as blank. A linear solvent gradient was applied in decreasing organic solvent (80–35%, 16 min) at 0.15 ml min⁻¹ flow rate and 45 °C column oven temperature. Mobile phases were aqueous 200 mmol per litre ammonium bicarbonate solution (pH 9.3, adjusted with 25% ammonium hydroxide), 100% acetonitrile and 100% water. Ammonium bicarbonate solution was kept at 10% throughout the run, resulting in a steady 20 mmol per litre concentration. Metabolites were analyzed using a mass spectrometer with a heated electrospray ionization source using polarity switching and the following settings: resolution of 70,000 at m/z of 200; spray voltages of 3400 V for positive and 3000 V for negative mode; sheath gas of 28 arbitrary units (AU) and auxiliary gas of 8 AU; vaporizer temperature of 280 °C; and ion transfer tube temperature of 300 °C. The instrument was controlled using Xcalibur 4.1.31.9 software (Thermo Scientific). Metabolite peaks were confirmed using commercial standards (Sigma-Aldrich). Data quality was monitored throughout the run using an in-house quality control cell line extracted in a similar manner to other samples.

Metabolite peaks areas were exported from TraceFinder 5.1. and analyzed in R (R Core Team, 2021) and Python. Peak areas were corrected by subtracting the areas observed in blank samples (extraction buffer) and normalized to the TMI (total metabolic intensity) calculated for each sample. Labeled fractions were obtained after combining the peak areas of the labeled (m + 1 to m + n) isotopologues. The differential abundance between genotypes was analyzed with limma[62] using lmFit with eBayes correction on the contrasts between genotypes, after replacing missing values with zeros while flagging the number of missing metabolites. Metabolites with FDR-corrected $p < 0.01$ were considered differentially abundant if they contained less than 3 missing values. Volcano plots were created with Python Matplotlib[63].

To study lactate metabolism, we did a similar tracing experiment using 2 mM U³C-lactate (Cambridge Isotope Laboratories # CLM-10768-PK) for 48 h based on[30]. To detect key metabolites, including acetyl-CoA and acetylcholine, we performed a targeted metabolic tracing experiment with U¹³C-glucose for 48 h labeling as before, but samples were processed differently as in ref. 64.

**Proteomics.** On day 30, control 0%(1) and mutants (52% and 54%) iPSC-MN were washed once with PBS, and then lysed in 6 M urea in 50 mM Tris-HCl, pH 8.0 and stored at −80 °C until analysis. Five separate wells were used as technical replicates.

Samples were reduced with 10 mM D,L-dithiothreitol and alkylated with 40 mM iodoacetamide and then digested overnight with sequencing grade modified trypsin (Promega). After digestion peptide, samples were desalted with a Sep-Pak tC18 96-well plate (Waters) and evaporated to dryness. Samples were dissolved in 0.1% formic acid and peptide concentration was determined with a NanoDrop device. The LC-ESI-MS/MS analysis was performed on a nanoflow HPLC system (Easy-nLC1200, ThermoFisher Scientific) coupled to the Orbitrap Fusion Lumos mass spectrometer (ThermoFisher Scientific) equipped with a nano-electrospray ionization source and FAIMS interface. Compensation voltages of −50 V and −70 V were used. Peptides were first loaded on a trapping column and subsequently separated inline on a 15 cm C18 column (75 μm × 15 cm, ReproSil-Pur 3 μm 120 Å C18-AQ, Dr. Maisch HPLC GmbH, Ammerbuch-Entringen). The mobile phase consisted of water with 0.1% formic acid (solvent A) or acetonitrile/water (80:20 (v/v)) with 0.1% formic acid (solvent B). A 120 min gradient was used to elute peptides (60 min from 5 to 21% solvent B followed by 48 min from 21 to 36% solvent B and in 6 min from 36 to 100% of solvent B, followed by 6 min wash stage with solvent B). Samples were analyzed by a data independent acquisition (DIA) LC-MS/MS method. Wash runs were submitted between each sample to reduce potential carry-over of peptides. MS data was acquired automatically by using Thermo Xcalibur 4.6 software (ThermoFisher Scientific). In a DIA method a duty cycle contained one full scan (400–1000 m/z) and 30 DIA MS/MS scans covering the mass range 400–1000 with variable width isolation windows.

Protein identifications was done using DirectDIA and label free quantifications were done with MaxLFQ using Spectronaut software (Biognosys; version 18.0.2) against Swiss-Prot (2023_01 Homo Sapiens) and Universal Protein Contaminant database[65]. Precursor and protein detection FDR cutoff was 0.01. Quantification was carried out at MS2 level with area under the curve within integration boundaries for each targeted ion. The protein peak areas were exported from Spectronaut and analysed in R (R Core Team, 2021). Missing values were replaced with zeros, while flagging the number of missing proteins. Protein peak areas were log1p-transformed and quantile normalized using voom[62]. The differential expression between genotypes was analysed in R (R Core Team, 2021) with limma[62] using lmFit with eBayes correction on the contrasts between genotypes. Proteins with FDR-corrected $p < 0.01$ were considered differentially expressed if they contained less than 3 missing values. Differentially expressed proteins were analysed for enrichment against Reactome, KEGG, DO, WP, and GO databases using clusterProfiler[66]. Figures were generated with R ggplot2[67], except for volcano plots that were generated with Python Matplotlib[63]. Mitochondrial proteome identifications were based on Mito-Carta 3.0[68].

## Statistics and reproducibility

Significant differences between more than two groups were analyzed with one-way ANOVA followed by Tukey´s multiple comparison post-hoc test unless otherwise indicated, using GraphPad Prism v9.5.1 (733) (GraphPad Prism 101.2. software). Mean ± standard deviation was reported in all graphs. P-values under 0.05 were considered statistically significant. Unless otherwise indicated, experiments were performed three times using independently differentiated iPSC-derived motor neurons ($n = 3$ independent experiments), each including at least three technical replicates (three wells per cell line). Certain experiments, such as high-throughput analyses, were conducted once, while others were validated using orthogonal methods, including measurements of acetylcholine levels.

## Data availability

Metabolomics raw data has been submitted to Metabolomics Workbench[69] (accession number: PR002167, https://doi.org/10.21228/M8HJ9G), and the normalized and blanked data as well as the subsequent analysis are available in Supplementary Data 1. Proteomics raw data has been deposited in MassIVE (accession number: MSV000095767) and can be found together with the analyzed data in Supplementary Data 2. Source data for all figures can be found in Supplementary Data 3. Uncropped and unedited blot/gel images can be found in the Supplementary Fig. 9.

## Code availability

R (version 4.5.1) and Python (version 3.11) code to process, analyze and visualize the high throughput data using standard pipelines is available upon request.

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

## Acknowledgements

We acknowledge the expert support by Miia Nissilä and Juhana Juutila. The services by the Biomedicum Stem Cell Center, FIMM Metabolomics, FIMM High content imaging and analysis unit (FIMM-HCA), and Turku Proteomics Facility (supported by Biocenter Finland) are acknowledged. This study has been supported by the Academy of Finland Centre of Excellence on Stem Cell Metabolism (MetaStem, #353008), Academy of Finland Postdoctoral fellowship (#340207) to R.T.-M., and Sigrid Juselius Foundation (#240225).

## Author contributions

Conceptualization, R.T-M. and H.T.; Methodology, R.T-M., J.T., R.E., S.H., J.K., and J.P.; Investigation, R.T-M., J.T., and H.T.; Writing – Original Draft, R.T-M., and H.T.; Writing – Review & Editing, all authors; Funding Acquisition, R.T-M., E.Y,. and H.T.; Resources, V.H.; Supervision, R.T-M. and H.T.

## Competing interests

The authors declare no competing interests.
