## [Transparent Peer Review file · Communications Biology]

Metabolic costs and trade-offs of hypermetabolism in human motor neurons with ATP synthase deficiency

Corresponding Author: Professor Henna Tyynismaa

Version 0:

Reviewer comments:

Reviewer #1

(Remarks to the Author)

The Authors use iPSC-MNs harboring ~50% heteroplasmy for a disease variant in ATP6 to show that these iPSC-MNs have increased O₂ consumption (unrelated to coupling efficiency) together with elevated ECAR. Alongside these observations are decreased pantothenate (not due to changes in transport expression) and a change in how acetyl-CoA is used, as well as broad changes in the mitochondrial proteome. These interesting observations are clearly presented. Furthermore, these observations address the paucity of information on how metabolism might change - and what the consequences could be - in the context of mitochondrial disease. Therefore, this nicely performed study can have an important contribution.

A major drawback of the study is that I do not think there is evidence for a mechanistic link between hypermetabolism and the reported changes in metabolism. In particular, the metabolic changes shown (decreased pantothenate and evidence for shifts in acetyl-CoA use) would not incur a net increase in demand for energy. Furthermore, while changes in coupling efficiency of oxphos can be ruled out a source of changed energy demand, there can be other key sources of energy that could change (eg, Na-K-ATPase activity, PMCA activity (ATP activity for Ca²⁺ extrusion)) that are not considered.

My suggestion would be to do some re-writing of the manuscript to remove the link between hypermetabolism and the changes in metabolism, and, instead, focus on the change in acetyl-CoA "fate". One could still start with the OCR and ECAR, but present those data as "broad metabolic characterization" that suggest broad changes in metabolism.

Experimental:

To better understand if ECAR indeed reflects glycolysis, it would be useful to determine the 2-DG-sensitive ECAR (ie., the ECAR that would be attributable to glycolysis).

For OCR and ECAR data, it would be useful to emphasize in Results and Legend that the data are normalized to DNA.

Reviewer #2

(Remarks to the Author)

The manuscript by Torregrosa-Munumer et al presents a well-written and well structured study focused on the metabolic and molecular consequences of heteroplasmic mutations in mt-ATP6, affecting ATP synthase assembly, in iPSC-derived motor neurons. The authors conclude that mitochondria in iPSC-MNs with ~50% heteroplasmy undergo proteomic remodeling to support the state of hypermetabolism, and demonstrate an increased demand for Acetyl CoA. Their conclusions are supported by a comprehensive set of confirmatory experiments, with most experiments performed in 2 independent iPSC clones with either 0% or ~50% heteroplasmy (4 clones total), adding robustness to their findings. The data are clearly presented, and the experimental design is thoughtful and rigorous throughout.

One major concern that requires addressing is the question of directionality. Is hypermetabolism truly the cause of the remodeling, or rather the consequence? The authors should clarify this point and acknowledge the current limitations in inferring causality and directionality. An argument may be made based on first principles, but this needs to be explicit.

A few minor points also require clarification or adjustment:

1. The introduction takes the reader on an in-depth molecular tour of the OxPhos and ATP synthase system. Is this needed to set the stage for the main point of the paper? Perhaps a more appropriate setting around evolved limits to energy expenditure would be more appropriate and help the reader appreciate why hypermetabolism may be a problem – an unsustainable metabolic state?
2. What was the differentiation efficiency of the iPSC-MN cultures? Was the culture purity comparable between control and mutant lines?
3. To support the claim of hypermetabolism, total ATP consumption should be measured or approximated, for instance using ATP production rates derived from OCR and ECAR (see work by De Sousa et al. PMID: 37548091)
4. Throughout the paper, the terms mitochondrial function and mitochondrial dysfunction are overly general and seem at odds with the recognized multi-functional nature of mitochondria. In many cases, the authors seem to mean “impaired OxPhos capacity”, but in others they mean something more general. The elephant paper by Monzel et al. (PMID 37100996) may help in addressing this point.
5. The results are almost completely devoid of quantitative measures. How much hypermetabolism? 10%, 50%, 100% (doubling), etc. This was striking across lines #119-132, but applies throughout. In humans, 10% above predicted is typically considered “hypermetabolism”, so although things could be different in cultured cells, the magnitude here is relevant.
6. For proteomics, was only one control (0%) clone used? Or is this the average of the two lines?
7. Figure 3A needs a legend for colors, or additional labels.
8. Figures 4H/I the scale bar should probably be centered on zero (0) so the color accurately reflect up and down regulation. Same for other heatmaps. Otherwise most readers will assume the orange reflects upregulation where it is only the absence of downregulation.
9. Please indicate how many metabolites were significantly UP vs DOWN regulated. Same for proteins. From first principles, just more upregulated proteins (and metabolites) has to incur an energetic cost, since the synthesis of proteins consumes significant ATP (4 ATP molecules just for polypeptide synthesis, plus all other costs of folding, PTM, degradation, etc).
10. This requires confirming that the proportion of the ECAR signal related to respiration and other sources. ECAR should be measured in the absence of glucose to establish the dependency of this signal on glucose and glycolysis.
11. Figure 1B – please add “sc = supercomplex” to the legend for clarity
12. Figure 4G - the color scale is not intuitive. I suggest reversing it to align Figures 4H, 4I, and 3C (lower = blue, higher = yellow), or to consider using a different color scheme altogether.
13. Can the authors speculate about whether hypermetabolism in neurons, secondarily to mitochondrial OxPhos defects, could contribute to brain hypermetabolism observed in early stage neurodegeneration (which later becomes hypometabolic)?

Congratulations to the authors on this beautiful and carefully executed study. This will become a valuable contribution to the field.

Reviewer #3

(Remarks to the Author)

Review ‘Metabolic Costs and trade-offs of hypermetabolism in human motor neurons with ATP synthase deficiency’.

I have read this manuscript with great interest and congratulate the authors with this amazing work. Studying the phenomenon of hypermetabolism in mitochondrial diseases is novel and highly relevant as it guides therapeutic options and outcomes. I have a few minor comments.

Introduction:

1. Page 3 line 55: ‘subsequent energy crisis in cells.’ Could the authors rephrase ‘Energy Crisis’? I feel that the wording is too strong for what is actually known about the pathophysiology of mitochondrial diseases. The authors could use: ‘Subsequent energy deficits in cells.’

Results:

1. Could the authors mention the location of the MT-ATP6 variant that was inserted in the first paragraph of the results section? The c. position.
2. Figure 1B line 866: the authors should mention what SC refers to in the figure legend (not just in the main text/results section).
3. Line 119: “...Seahorse FX analyzer...” should be Seahorse XF analyzer.
4. Figure 3A: The authors should indicate in the figure legend what the cutoff was for differentially expressed proteins (p < ...?).

5. Figure 3A: There are a lot of proteins differentially expressed between the two heteroplasmy conditions. It would be interesting to zoom in on these. Do the authors consider this experimental/technical variation? What kind of proteins are they and which pathways are these differentially expressed proteins involved in?

Discussion

1. The authors suggest that avanafil might not work in the patients with MT-ATP6 mutations included in this manuscript due to the fact that their membrane potential was unaltered:

“This genotypic specificity may also explain why avanafil treatment, which successfully rescued an MT-ATP6 missense variant that reduced mitochondrial membrane potential, did not alleviate the hypermetabolic phenotype in our MT-ATP6 nonsense mutant neurons.”

I feel that this statement might be a bit preliminary. The authors solely test levels of glucose incorporation into lactate and citrate levels after avanafil treatment. It would be interesting to test the effects of avanafil on multiple aspects of mitochondrial functioning, including basal resp. capacity, proton leak, pantoate levels, etc. before definite conclusions can be drawn about how and why avanafil does not change lactate and citrate levels. I suggest that the authors either remove the aforementioned sentence or rephrase it to something along the lines of: “Avanafil did not alter levels of glucose incorporation into lactate or citrate, suggesting that this drug cannot change the metabolic changes observed in our MT-ATP6 nonsense mutant.”

Figure legends:

1. Figure 1: For Figure 1G, three data points are shown for most cell lines, while for the 0% heteroplasmy (2) only one data point is shown – but still an error bar is present. Where is the error bar based on if there is only one data point? In addition, the authors state that for this experiment, n= 3 independent experiments with 2 to 11 wells per cell line per experiment were used. Could the authors specify in the figure legend what the dots represent? Different wells in one representative experiment? Averages?

2. Figure 3C: The Y-axis label “selected upregulated proteins” seems to be cropped incorrectly.

3. The labelling of Figure 4 starts with (C). Should be (A)?

4. Supplementary Figure 1D. What do the dots represent? “The data represents n=3 independent differentiation experiments, each with two independent wells measured by duplicate.” The two independent wells or three independent experiments?

Version 1:

Reviewer comments:

Reviewer #1

(Remarks to the Author)

Thank you for your very clear replies to my comments.

I think the manuscript is improved, especially with the addition of the calculation of ATP synthesis rate using the Desouza et al 2023 method, and better use of the ECAR measurement to reflect glycolysis.

I also think that focusing the Introduction to reflect hypermetabolism was a great idea. In that regard, I would suggest either adding a little more or reshaping a little the introduction to reflect that 1) elevated energy expenditure can be deemed “hypermetabolism” with as little as 10% rise in resting energy expenditure (PMID: 36009527; PMID: 29706605), and 2) can be caused by increased ATP turnover (reflecting great ATP demand, for which there can be many sources (PMID: 9234964)) or increased uncoupling of oxidation from phosphorylation as occurs in Luft’s Disease. I think the latter provides a more general framework for hypermetabolism.

Also, line 72 2nd page of Introduction: Should not “integrated stress responses” be the “integrated stress response” or “stress responses”?

Congratulations on this excellent work that I expect to be a useful piece of the literature.

Reviewer #2

(Remarks to the Author)

All comments have been satisfactorily addressed.

The additions of calculated ATP synthesis rates add substantially to the impact of the paper. Excellent work! Kudos for performing the measurements across multiple cell lines. This is a beautiful paper.

Reviewer #3

(Remarks to the Author)

The authors have answered my questions in a satisfactory manner.

Reviewer #1 (Remarks to the Author):

The Authors use iPSC-MNs harboring ~50% heteroplasmy for a disease variant in ATP6 to show that these iPSC-MNs have increased O₂ consumption (unrelated to coupling efficiency) together with elevated ECAR. Alongside these observations are decreased pantothenate (not due to changes in transport expression) and a change in how acetyl-CoA is used, as well as broad changes in the mitochondrial proteome. These interesting observations are clearly presented. Furthermore, these observations address the paucity of information on how metabolism might change- and what the consequences could be- in the context of mitochondrial disease. Therefore, this nicely performed study can have an important contribution.

We appreciate these insightful comments. We have addressed each of them below and have revised the manuscript accordingly. All changes in the manuscript text are highlighted in yellow.

A major drawback of the study is that I do not think there is evidence for a mechanistic link between hypermetabolism and the reported changes in metabolism. In particular, the metabolic changes shown (decreased pantothenate and evidence for shifts in acetyl-CoA use) would not incur a net increase in demand for energy. Furthermore, while changes in coupling efficiency of oxphos can be ruled out a source of changed energy demand, there can be other key sources of energy that could change (eg, Na-K-ATPase activity, PMCA activity (ATP activity for Ca²⁺ extrusion)) that are not considered. My suggestion would be to do some re-writing of the manuscript to remove the link between hypermetabolism and the changes in metabolism, and, instead, focus on the change in acetyl-CoA "fate". One could still start with the OCR and ECAR, but present those data as "broad metabolic characterization" that suggest broad changes in metabolism.

We appreciate these concerns. In response, we have added new evidence that strengthens the link between the MT-ATP6 mutation and hypermetabolism. The term "hypermetabolism" has been used in various ways over the past several decades; however, in the context of mitochondrial diseases, it is now recognized as a state of reduced metabolic efficiency coupled with elevated energy expenditure (Sturm et al, 2023. PMID: 36635485). Guided by this comprehensive study and the recommendation of Reviewer #2, we re-analyzed our Seahorse XF data and computed the total ATP production rate following DeSousa et al. (PMID: 37548091) work. This approach allowed us to estimate the ATP generated via both OxPhos and glycolysis. The sum of these values (the total ATP flux) captures the overall energy demand of each cell line, and it is equivalent to the resting energy expenditure measured in humans and mice through indirect calorimetry (Sturm et al, 2023. PMID: 36635485).

The total ATP flux (ATP generated via OxPhos + glycolysis) revealed that iPSC-derived MN with ~50 % MT-ATP6 heteroplasmy show, on average, a ~35 % increase in energy demand (104.2 pmol ATP/min/total DNA) compared with control neurons (76.97 pmol ATP/min/total DNA). This increased demand is comparable to the ~30 % overall increase in resting whole-body oxygen consumption measured by indirect calorimetry and expressed relative to body weight (used as an estimate of resting energy expenditure) in patients carrying pathogenic mitochondrial mutations (Sturm et al., 2023; PMID 36635485).

[redacted]

Based on our results, we think the term "hypermetabolism" is appropriate: in the mutant motor neurons the metabolism is altered with increased ATP production rates. However, we have revised the manuscript to avoid implying a direct causal link between the higher ATP production rates and acetyl-CoA utilization. We acknowledge that mechanistically linking the increased ATP demand to altered acetyl-CoA fate is challenging, especially because mitochondrial pyruvate oxidation appears unchanged. Nonetheless, a defining aspect of hypermetabolism is the broad metabolic rewiring that can generate futile cycles, wasting both substrates and energy, which aligns with our study. Overall, we consider our findings significant because they provide the first quantitative demonstration of hypermetabolism in motor neurons harboring a pathogenic mtDNA variant, establishing a strong foundation for future research.

Experimental:

1. To better understand if ECAR indeed reflects glycolysis, it would be useful to determine the 2-DG-sensitive ECAR (ie., the ECAR that would be attributable to glycolysis).

To clarify the origin of the elevated ECAR, we set-up and performed an assay using Seahorse XF, the **Glycolytic Rate Assay**, which provides a specific quantification of the glycolytic contribution to proton efflux (PER) following two injections with rotenone + antimycin and then 2-DG. We calculated the buffering factor of our assay media, and Seahorse XF values were normalized to DNA per well. The 2-DG-sensitive ECAR (basal ECAR – post-2-DG-ECAR) and GlycoPER (glycolytic contribution to PER, derived from the previous calculation) were elevated in mutant motor neurons (New Figure 1I), although the statistically significant differences ($p < 0.05$) compared with the control line 0%(2) were limited. Moreover, our previous [U-¹³C]-glucose tracing experiment revealed a robust increase in fully labelled lactate (Figure 2D), consistent with greater cumulative glycolytic throughput. Together, we trust that **these complementary approaches support the conclusion that the elevated ECAR in mutant neurons reflects increased glycolysis.**

Based on these new results, we have revised the manuscript to include the results of the Seahorse XF Glycolytic Rate Assay: glycolytic rate (GlycoPER) as new **Figure 1I**; and Proton Efflux Rate kinetics and Compensatory Glycolysis as new **Supplementary Figure 1J** and **K** respectively. The corresponding text in the manuscript has been updated accordingly (lines 124-132). Compensatory glycolysis following rotenone/antimycin injection did not differ between groups, indicating that mutant and control neurons mount a similar glycolytic response under mitochondrial stress, and therefore, glycolysis is not impaired.

2. For OCR and ECAR data, it would be useful to emphasize in Results and Legend that the data are normalized to DNA.

Thank you for the suggestion. To make it clearer, we added (line 964): "Data was normalized to total DNA per well" in the legend of Figure 1, and also in Sup. Fig. 1. In the Results section, we included (lines 118-119): "We used a Seahorse XF analyzer normalized to total DNA (proxy of cell number) to measure the oxygen consumption rate (OCR)...".

Reviewer #2 (Remarks to the Author):

The manuscript by Torregrosa-Munumer et al presents a well-written and well structured study focused on the metabolic and molecular consequences of heteroplasmic mutations in mt-ATP6, affecting ATP synthase assembly, in iPSC-derived motor neurons. The authors conclude that mitochondria in iPSC-MNs with ~50% heteroplasmy undergo proteomic remodeling to support the state of hypermetabolism, and demonstrate an increased demand for Acetyl CoA. Their conclusions are supported by a comprehensive set of confirmatory experiments, with most experiments performed in 2 independent iPSC clones with either 0% or ~50% heteroplasmy (4 clones total), adding robustness to their findings. The data are clearly presented, and the experimental design is thoughtful and rigorous throughout.

We appreciate these insightful comments. We have addressed each of them below and have revised the manuscript accordingly. All changes in the manuscript text are highlighted in yellow.

One major concern that requires addressing is the question of directionality. Is hypermetabolism truly the cause of the remodeling, or rather the consequence? The authors should clarify this point and acknowledge the current limitations in inferring causality and directionality. An argument may be made based on first principles, but this needs to be explicit.

Determining the directionality of these processes is indeed challenging, and we have attempted to address this complexity in the final figure of our manuscript, where we presented our running hypothesis or model. Impaired ATP synthase function resulting from the MT-ATP6 mutation is the initial trigger. As maturing neurons depend heavily on oxidative phosphorylation, they may experience pressure to optimize ATP production and maintain the mitochondrial membrane potential, potentially driving the observed mitochondrial proteomic remodeling. This remodeling extends beyond the mitochondria, influencing other cellular processes such as histone acetylation, which may modulate or intensify the hypermetabolic phenotype. While hypermetabolism might primarily result from the proteomic remodeling, it could also generate new pressures that drive further proteomic changes. In essence, we think that these processes are interdependent and evolve together, reflecting a dynamic interplay rather than a simple cause-and-effect relationship.

We acknowledge the current limitations in establishing causality and directionality and appreciate your insights on this matter. We have added the following text in the discussion (lines 407-413):

Our findings indicate that iPSC-MNs with ~50% heteroplasmy undergo significant proteomic changes to sustain a hypermetabolic state. Determining the causal direction of these changes, however, is challenging. While the proteomic remodeling may drive hypermetabolism, it is also possible that the hypermetabolic state imposes new demands that trigger additional proteomic adaptations. This would imply a reciprocal relationship, where both processes influence each other in a continuous, dynamic feedback loop rather than following a linear cause-and-effect model.

A few minor points also require clarification or adjustment:

1. The introduction takes the reader on an in-depth molecular tour of the OxPhos and ATP synthase system. Is this needed to set the stage for the main point of the paper? Perhaps a more appropriate setting around evolved limits to energy expenditure would be more appropriate and help the reader appreciate why hypermetabolism may be a problem – an unsustainable metabolic state?

We have slightly revised the introduction to reduce the emphasis on OxPhos and ETC function and provide a broader context for the concept of energy constraints, balance, and an unsustainable metabolic state.

2. What was the differentiation efficiency of the iPSC-MN cultures? Was the culture purity comparable between control and mutant lines?

Initially, we only included some representative images of ISL1/2- or Hb9 stainings as prove of the motor neuron differentiation (Figure 1C, lower panels). Now, we validated motor neuron differentiation efficiency using these two widely established motor neuron markers by quantifying the percentage of positive cells relative to total nuclei (DAPI-positive). Quantification was performed using an automated and unbiased ImageJ-based analysis of ~1,000 cells and added as new Figures 1D and E. The differentiation efficiency was comparable among control and mutant cell lines.

3. To support the claim of hypermetabolism, total ATP consumption should be measured or approximated, for instance using ATP production rates derived from OCR and ECAR (see work by De Sousa et al. PMID: 37548091).

Thank you for the suggestion. We estimated the OxPhos and glycolytic ATP production rates following the method described by De Sousa et al., after determining the buffer capacity of our media and experimental conditions. Non-mitochondrial respiration was subtracted from ATP OxPhos estimates to avoid attributing hypermetabolism to non-mitochondrial sources and the data was normalized to total DNA per well. We found that total ATP production rate (ATP produced through OxPhos and glycolysis) was overall increased in mutant iPSC-MN. The 54% mutant iPSC-MNs vs. 0%(2) comparison was near statistical significance ($p = 0.067$). We have modified **Figure 1**, adding total ATP production rate as the new **Figure 1J** and modified the first section of the results accordingly (lines 133-149).

4. Throughout the paper, the terms mitochondrial function and mitochondrial dysfunction are overly general and seem at odds with the recognized multi-functional nature of mitochondria. In many cases, the authors seem to mean “impaired OxPhos capacity”, but in others they mean something more general. The elephant paper by Monzel et al. (PMID 37100996) may help in addressing this point.

We fully agree and have revised the text to be more specific whenever referring to mitochondrial function or dysfunction, following Monzel et al. paper.

5. The results are almost completely devoid of quantitative measures. How much hypermetabolism? 10%, 50%, 100% (doubling), etc. This was striking across lines #119-132, but applies throughout. In humans, 10% above predicted is typically considered “hypermetabolism”, so although things could be different in cultured cells, the magnitude here is relevant.

Thank you very much for raising this important point. Based on calculations of ATP production rates as described by De Sousa et al. (PMID: 37548091), and guided by Sturm et al, 2023. (PMID: 36635485), we have now computed the total ATP production rate (ATP OxPhos + ATP glycolysis). Our analysis revealed that iPSC-MNs with ~50% heteroplasmy exhibited a ~35% increase on average in total ATP demand / energy expenditure per minute (76,97 total pmol ATP/min/DNA for the control motor neurons, and 104,2 total pmol ATP/min/DNA for mutant motor neurons), closely mirroring the ~30% elevated resting energy expenditure observed in patients with mtDNA pathogenic variants (Sturm et al, 2023. PMID: 36635485) and determined through indirect calorimetry. This increase in mutants affects both OxPhos and glycolysis similarly, as mutant motor neurons retain a comparable reliance on each pathway to that of controls (approximately 90% of ATP is derived from OxPhos and 10% from glycolysis).

We have added four new figures to the manuscript:

- **Figure 1J**, Total ATP production rate (ATP OxPhos + ATP glycolysis), as described in the #Reviewer 2 comment number 3.
- **Sup. Fig. 1L**, ATP production as in Figure 1J, with the contributions from OxPhos and glycolysis displayed separately.
- **Sup. Fig. 1M**, numerical data from Figure 1J to more clearly illustrate the ~35% increase in total ATP production rate observed in mutant iPSC-MNs
- **Sup. Fig. 1N**, % of total ATP. To highlight the relative contributions of each pathway, we expressed OxPhos- and glycolysis-derived ATP production as a percentage of total ATP production. In both control and mutant iPSC-MNs, approximately 90% of ATP was derived from mitochondrial respiration and 10% from glycolysis. These ratios were maintained between groups, indicating that the increased ATP production rates observed in mutant cells was proportionally elevated across both pathways, without a clear shift in metabolic preference.

We have also modified the first section of the results text accordingly (lines 133-146).

6. For proteomics, was only one control (0%) clone used? Or is this the average of the two lines?

Due to economic constraints, proteomic analysis was performed exclusively on the first control group (0%(1)). We indicated this in the material and methods section, but for the shake of clarity, we have now also indicated it in the **Figure 3** legend (lines 995-997):

(...) (A) Venn diagram reveals a large amount of shared differently expressed protein between mutants and the **0%(1)** (755).
 (B) Pathway Enrichment Analysis of Reactome terms enriched pathways in MUT (52% + 54%) vs WT (**0%(1)**). (...)

7. Figure 3A needs a legend for colors, or additional labels.

We have modified the Venn diagram to make it clearer. We highlighted in orange the region representing the shared differently expressed proteins in mutants (52 and 54%) vs control (0%).

8. Figures 4H/I the scale bar should probably be centered on zero (0) so the color accurately reflect up and down regulation. Same for other heatmaps. Otherwise, most readers will assume the orange reflects upregulation where it is only the absence of downregulation.

We have updated all the heatmaps (**Figures 3C** and **4H/I**) to center the color scale at 0, with grey indicating no change ($\log_2FC = 0$), blue representing downregulation ($\log_2FC < 0$), and orange indicating upregulation ($\log_2FC > 0$). Additionally, we have reversed the gene order in **Figures 4H** and **4I** so that the most downregulated genes now appear at the bottom of the heatmaps.

9. Please indicate how many metabolites were significantly UP vs DOWN regulated. Same for proteins. From first principles, just more upregulated proteins (and metabolites) has to incur an energetic cost, since the synthesis of proteins consumes significant ATP (4 ATP molecules just for polypeptide synthesis, plus all other costs of folding, PTM, degradation, etc).

		Down	Up
Proteomics			
MUT (52+54%) vs WT (0%)		756	599
Metabolomics			
MUT (52+54%) vs WT (0%(1) and (2))	1h	1	7
	12h	3	25
	48h	14	54

Metabolites showed a consistent increase in the mutants, particularly after 48 hours of [U¹³C]-glucose exposure, which was accompanied by a decrease in lower isotopologue counterparts. In the pooled samples (MUT versus WT), we observed substantial changes in protein abundance. Although downregulated proteins predominated in the mutants, with approximately 150 more downregulated than upregulated, the overall numbers of differentially expressed proteins were broadly similar (about 600 down and 750 up).

We appreciate the reviewer's thoughtful comment. While simple counts of up- vs downregulated metabolites or proteins can be provided, we believe they may not be the most informative measure of the biological processes at play. The energetic impact of proteome and metabolome remodelling is unlikely to depend solely on the number of molecules up- or downregulated, but also on their functional categories, turnover rates, and pathway-level integration. For this reason, we have chosen to emphasize pathway enrichment analysis and functional interpretation in the manuscript, which we believe more accurately reflects the biological significance of the observed changes.

10. This requires confirming that the proportion of the ECAR signal related to respiration and other sources. ECAR should be measured in the absence of glucose to establish the dependency of this signal on glucose and glycolysis.

To clarify the origin of the elevated ECAR, we optimized the conditions and performed complementary Seahorse XF analyses in iPSC-derived motor neurons.

Neurons are post-mitotic cells and meet most of their energetic demands through OxPhos (for example, according to our own study, up to ~90% of ATP; see **New Supp. Fig. 1N**). This contrasts with proliferating cells, which are primarily glycolytic and produce large amounts of lactate to sustain the glycolytic flux. The strong reliance of neurons on mitochondria may limit the ECAR readout in the absence of glucose, as neurons are highly sensitive to glucose deprivation, which remains their main substrate for OxPhos.

We conducted the Seahorse XF Glycolysis Stress Test (n = 3 independent experiments) **to assess ECAR in the absence of glucose**. Cells were cultured in our regular differentiation medium (with 21.25 mM glucose) and were starved prior to the assay, when the culture medium was replaced with Seahorse assay media without glucose or pyruvate. ECAR values were normalized to DNA per well. We observed that ECAR before glucose injection (first three measurements) was higher in both control and mutant cells compared to ECAR after 2-DG treatment. *Non-glycolytic acidification* (last measurement before glucose addition) was slightly, although significantly inconsistently, increased in mutants. However, when we adjusted for non-glycolytic acidification by subtracting post-2-DG ECAR (ECAR without glucose – ECAR after 2-DG), no differences were observed between groups (see below). Similar results were obtained in motor neurons cultured in 5.5 mM glucose for 48 hours prior to the assay (data not shown). The cells were gently washed, and the differentiation medium was fully replaced with glucose-free assay media before the Seahorse XF run. Nevertheless, the observed elevated ECAR in the apparent absence of glucose (higher than the post-2DG measurements) in all cell lines likely reflects ongoing glycolysis, either fueled by residual glucose in the media or by intracellular glucose stores. Since neurons are highly sensitive to glucose deprivation and the neuronal network is delicate to media changes and more thorough washes, technical limitations prevent us from obtaining a clearer readout.

Glycolysis Stress Test, only for the review process and not included in the revised manuscript.

[redacted]

However, we performed another Seahorse XF assay, the Glycolytic Rate Assay (n = 2 independent experiments), which provides a more specific **quantification of the glycolytic and mitochondrial contribution to ECAR**. The buffering factor for our assay media was calculated ad hoc, and values were normalized to DNA per well. The 2-DG-sensitive ECAR (basal ECAR – post-2-DG-ECAR) and GlycoPER (glycolytic contribution to PER, derived from the previous calculation) were elevated in mutant motor neurons (new **Figure 1I**), although the statistically significant differences (p < 0.05) compared with the control line 0%(2) were limited. Furthermore, the mitochondrial contribution to proton efflux (mitoPER) accounted for only ~1% of total PER (Proton Efflux Rate, see below). **These results suggest that ECAR is primarily explained by changes in glycolysis, with a very limited mitochondrial contribution**. We also observed that compensatory glycolysis following

rotenone/antimycin injection did not differ between groups, indicating that mutant and control neurons mount a similar glycolytic response under mitochondrial stress, and therefore, glycolysis is not impaired.

Moreover, our 48-hour metabolomics experiments capture cumulative effects over time, providing a more integrated view of cellular metabolism. Our metabolomics experiments revealed a robust increase in fully labelled lactate (**Figure 2D**), consistent with greater overall glycolytic throughput. Therefore, **we believe that these complementary approaches support the conclusion that the elevated ECAR in mutant neurons primarily reflects increased glycolysis.**

Based on these new results, we have revised the manuscript to include the results of the Seahorse XF Glycolytic Rate Assay: glycolytic rate (GlycoPER) as **Figure 1I**; and Proton Efflux Rate kinetics and Compensatory Glycolysis as **Supplementary Figure 1J** and **K** respectively. The corresponding text in the manuscript has been updated accordingly (lines 124-132).

Seahorse Glycolysis Rate Assay

11. Figure 1B – please add “sc = supercomplex” to the legend for clarity.

Thank you for the suggestion. We added to the legend of Figure1B the meaning of “sc” (lines 945-949):

*(B) Representative cropped immunoblot of Complex V assembly determined by Blue-Native PAGE. Complex V (CV) was detected using anti-ATP5FA1, targeting the F1 subunit (facing the matrix) of ATP synthase; and Complex II (CII) was detected using anti-SDHA, which served as the loading control. n = 2 technical replicates (independent wells). **sc = subcomplexes.** (...)*

12. Figure 4G- the color scale is not intuitive. I suggest reversing it to align Figures 4H, 4I, and 3C (lower = blue, higher = yellow), or to consider using a different color scheme altogether.

We have changed the color code of both enrichment pathway analysis figures (Figures 3B and 4G). Now, we used orange shades in Figure 3B (upregulation of pathways) and blue shades in Figure 4G (downregulation of pathways), with grey representing the smaller changes and orange or blue the biggest. We hope this new coloring makes the figures and their meaning clearer:

13. Can the authors speculate about whether hypermetabolism in neurons, secondarily to mitochondrial OxPhos defects, could contribute to brain hypermetabolism observed in early-stage neurodegeneration (which later becomes hypometabolic)?

Speculatively, mitochondrial OxPhos defects may force neurons to upregulate glycolysis and TCA cycle flux in an effort to maintain ATP production, potentially leading to a hypermetabolic state. However, the constraints involved may extend beyond bioenergetics alone. Impaired mitochondrial OxPhos can end up disrupting other critical mitochondrial functions, including calcium buffering and anabolic reactions involved in the synthesis of neurotransmitters such as acetyl choline, glutamate or GABA. These, along with ATP production, are crucial for neuronal excitability and synaptic function. Moreover, disruption of mitochondrial calcium handling can lead to elevated intracellular calcium, which increases neuronal excitability and, consequently, energy demands. These (mal)adaptive responses might temporarily preserve neuronal activity and network function. However, as mitochondrial metabolism and calcium buffering dysfunction progresses and compensatory mechanisms become overwhelmed, neurons may begin to retract synapses, lose connectivity, or undergo cell death, ultimately shifting the metabolic profile toward hypometabolism. This progression could reflect the loss of neuronal function and viability as the disease advances. However, this picture might be more complicated, as the brain and peripheral nerves contain many non-neuronal cell types (e.g., glial cells, muscle cells), whose diverse metabolic and functional interactions may be disrupted in various ways, which in turn, can also affect the neurons.

This is a very interesting question. But at this stage, we can only speculate, and that is why it has not been addressed in the manuscript. Further studies will be required to explore it.

Congratulations to the authors on this beautiful and carefully executed study. This will become a valuable contribution to the field.

Thank you very much for your encouraging feedback! We truly appreciate your constructive comments, which we believe have helped us improve both the quality and impact of our study.

Reviewer #3 (Remarks to the Author):

Review 'Metabolic Costs and trade-offs of hypermetabolism in human motor neurons with ATP synthase deficiency'.

I have read this manuscript with great interest and congratulate the authors with this amazing work. Studying the phenomenon of hypermetabolism in mitochondrial diseases is novel and highly relevant as it guides therapeutic options and outcomes. I have a few minor comments.

We appreciate these insightful comments. We have addressed each of them below and have revised the manuscript accordingly. All changes in the manuscript are highlighted in yellow.

Introduction

1. Page 3 line 55: 'subsequent energy crisis in cells.' Could the authors rephrase 'Energy Crisis'? I feel that the wording is too strong for what is actually known about the pathophysiology of mitochondrial diseases. The authors could use: 'Subsequent energy deficits in cells.'

Thank you for the recommendation. We have rephrased it as suggested.

Results

1. Could the authors mention the location of the MT-ATP6 variant that was inserted in the first paragraph of the results section? The c. position.

We added the location of the variant in the first paragraph of the results section (lines 97-100):

The studied motor neurons carried a 49% heteroplasmic nonsense variant (mitochondrial mutation load) in MT-ATP6 (m.9154C>T, p.Gln210Ter), which encodes the "α" subunit of the F₀ domain of the ATP synthase, forming the proton re-entry channel.

2. Figure 1B line 866: the authors should mention what SC refers to in the figure legend (not just in the main text/results section).

We added to the legend of Figure 1B the meaning of "sc" (lines 945-949):

*(B) Representative cropped immunoblot of Complex V assembly determined by Blue-Native PAGE. Complex V (CV) was detected using anti-ATP5FA1, targeting the F1 subunit (facing the matrix) of ATP synthase; and Complex II (CII) was detected using anti-SDHA, which served as the loading control. n = 2 technical replicates (independent wells). **sc = subcomplexes.** (...)*

3. Line 119: "...Seahorse FX analyzer..." should be Seahorse XF analyzer.

Thank you for spotting the typo. We have corrected it.

4. Figure 3A: The authors should indicate in the figure legend what the cutoff was for differentially expressed proteins ($p < \dots$).

We modified the Figure 3 legend as follows (lines 995-997):

Figure 3. The mitochondrial proteome in neurons is reorganized to support the hypermetabolic state. (A) Venn diagram reveals a large amount of shared differentially expressed protein between mutants and the 0%(1) (755). Proteins with FDR-corrected p-values < 0.01 were considered differentially expressed. (...)

5. Figure 3A: **There are a lot of proteins differentially expressed between the two heteroplasmy conditions.** It would be interesting to zoom in on these. Do the authors consider this experimental/technical variation? What kind of proteins are they and which pathways are these differentially expressed proteins involved in?

These differences can indeed reflect biological effects of heteroplasmy levels. The commonly cited threshold for biochemical or phenotypic effects from pathogenic mtDNA mutations is around 60%–80% heteroplasmy. However, this threshold is not absolute; the exact point at which functional changes occur can vary depending on the specific mutation, tissue, and cellular context (PMID: **28415858**). Even the same heteroplasmy levels of the same mutation in the same tissue can have different outcomes in different patients. It is still unclear whether cellular responses to increasing heteroplasmy are gradual or sharp, and different features (such as enzyme activity or gene expression) may respond at different levels of heteroplasmy. Because 52% and 54% heteroplasmy are so close (and both fall below the typical threshold for major functional effects) we treated these clones as biological replicates. However, we acknowledge that even a 2% difference in heteroplasmy could potentially have functional consequences. To minimize the risk of overinterpreting subtle differences, we focused our analysis on the changes that were consistently shared between the two clones.

Yet, when comparing alone the two mutant cell lines, 1514 proteins showed distinct significant changes: 670 were downregulated and 844 were upregulated in 54% heteroplasmy iPSC-MN compared to the 52%. Pathway enrichment analysis (Reactome) revealed overlap with some of the key pathways affected when pooling both mutants (52% + 54%) and comparing them to wild-type (0%), suggesting an amplification of these effects. In particular, we observed upregulation of pathways involved in Complex I biogenesis, respiratory electron transport, and broader metabolic processes in the 54% iPSC-MNs. Downregulated pathways were predominantly related to cell–matrix interactions, but also included neuronal functions and synaptic signaling—such as MECP2-regulated pathways, neuronal receptors, and synaptic protein interactions—indicating potential alterations in cell communication.

In summary, although the heteroplasmy difference between 52% and 54% is small and both levels fall below the typical threshold for mtDNA pathogenic variants, we observed a large number of differentially expressed proteins. This suggests that even minor shifts in heteroplasmy may elicit broader cellular responses. However, to avoid overinterpretation of clone-specific effects, we focused our analysis on the molecular changes that were consistent across both clones, aiming to highlight the most robust and reproducible heteroplasmy-associated alterations.

Top 15 downregulated and upregulated proteins (based on p-value) in 54% motor neurons versus 52%:

Top 25 upregulated and downregulated (based on Reactome) in 54% iPSC-MN versus 52%:

Upregulated in 54%						
Pathway identifi	Pathway name	# found	# total	ratio	pValue	FDR
R-HSA-6799198	Complex I biogenesis	13	71	0,0045	4,61E-04	2,05E-01
R-HSA-6790901	rRNA modification in the nucleus and cytosol	13	71	0,0045	4,61E-04	2,05E-01
R-HSA-5617472	Activation of anterior HOX genes in hindbrain development during early embryogenesis	17	116	0,0073	8,33E-04	2,05E-01
R-HSA-5619507	Activation of HOX genes during differentiation	17	116	0,0073	8,33E-04	2,05E-01
R-HSA-9933937	Formation of the canonical BAF (cBAF) complex	6	18	0,0011	8,61E-04	2,05E-01
R-HSA-9934037	Formation of neuronal progenitor and neuronal BAF (npBAF and nBAF)	7	25	0,0016	9,00E-04	2,05E-01
R-HSA-9932451	SWI/SNF chromatin remodelers	7	31	0,0020	2,99E-03	5,12E-01
R-HSA-9932444	ATP-dependent chromatin remodelers	7	31	0,0020	2,99E-03	5,12E-01
R-HSA-6794361	Neurexins and neuroligins	10	60	0,0038	3,88E-03	5,20E-01
R-HSA-9933947	Formation of the non-canonical BAF (ncBAF) complex	5	17	0,0011	3,91E-03	5,20E-01
R-HSA-445095	Interaction between L1 and Ankyrins	7	33	0,0021	4,19E-03	5,20E-01
R-HSA-6794362	Protein-protein interactions at synapses	13	93	0,0059	4,76E-03	5,43E-01
R-HSA-9845323	Regulation of endogenous retroelements by Piwi-interacting RNAs (piRNAs)	10	68	0,0043	8,94E-03	7,48E-01
R-HSA-447038	NrCAM interactions	3	7	0,0004	8,96E-03	7,48E-01
R-HSA-70895	Branched-chain amino acid catabolism	9	59	0,0037	1,03E-02	7,48E-01
R-HSA-210747	Regulation of gene expression in early pancreatic precursor cells	4	14	0,0009	1,06E-02	7,48E-01
R-HSA-9837999	Mitochondrial protein degradation	13	104	0,0066	1,14E-02	7,48E-01
R-HSA-3214842	HDMs demethylate histones	6	31	0,0020	1,18E-02	7,48E-01
R-HSA-9033500	TYSND1 cleaves peroxisomal proteins	3	8	0,0005	1,28E-02	7,48E-01
R-HSA-9865118	Diseases of branched-chain amino acid catabolism	6	32	0,0020	1,36E-02	7,48E-01
R-HSA-381038	XBP1(S) activates chaperone genes	12	95	0,0060	1,36E-02	7,48E-01
R-HSA-9841251	Mitochondrial unfolded protein response (UPRmt)	6	33	0,0021	1,56E-02	7,48E-01
R-HSA-9824585	Regulation of MITF-M-dependent genes involved in pigmentation	8	53	0,0033	1,59E-02	7,48E-01
R-HSA-9609507	Protein localization	18	171	0,0108	1,70E-02	7,48E-01
R-HSA-611105	Respiratory electron transport	20	197	0,0124	1,74E-02	7,48E-01

Downregulated in 52%						
Pathway identifi	Pathway name	# found	# total	ratio	pValue	FDR
R-HSA-8986944	Transcriptional Regulation by MECP2	22	100	0,0063	1,7E-08	2,3E-05
R-HSA-9022699	MECP2 regulates neuronal receptors and channels	12	32	0,0020	1,4E-07	9,3E-05
R-HSA-2426168	Activation of gene expression by SREBF (SREBP)	17	71	0,0045	2,2E-07	1,0E-04
R-HSA-6794362	Protein-protein interactions at synapses	19	93	0,0059	4,7E-07	1,6E-04
R-HSA-1655829	Regulation of cholesterol biosynthesis by SREBP (SREBF)	18	87	0,0055	7,9E-07	2,2E-04
R-HSA-3000157	Laminin interactions	10	31	0,0020	5,5E-06	1,3E-03
R-HSA-446353	Cell-extracellular matrix interactions	7	19	0,0012	6,2E-05	1,2E-02
R-HSA-6794361	Neurexins and neuroligins	12	60	0,0038	7,2E-05	1,2E-02
R-HSA-111465	Apoptotic cleavage of cellular proteins	9	38	0,0024	1,7E-04	2,5E-02
R-HSA-6806834	Signaling by MET	14	88	0,0056	2,0E-04	2,8E-02
R-HSA-8874081	MET activates PTK2 signaling	8	32	0,0020	2,6E-04	3,3E-02
R-HSA-428542	Regulation of commissural axon pathfinding by SLIT and ROBO	5	12	0,0008	4,0E-04	4,0E-02
R-HSA-112316	Neuronal System	43	490	0,0310	4,1E-04	4,0E-02
R-HSA-397014	Muscle contraction	25	232	0,0147	4,2E-04	4,0E-02
R-HSA-438066	Unblocking of NMDA receptors, glutamate binding and activation	7	27	0,0017	5,1E-04	4,6E-02
R-HSA-8875878	MET promotes cell motility	9	45	0,0028	5,6E-04	4,6E-02
R-HSA-428359	Insulin-like Growth Factor-2 mRNA Binding Proteins (IGF2BPs/IMPs/VICKZs) bind RNA	5	13	0,0008	5,7E-04	4,6E-02
R-HSA-9013423	RAC3 GTPase cycle	14	100	0,0063	7,0E-04	5,4E-02
R-HSA-9013408	RHOG GTPase cycle	12	78	0,0049	7,5E-04	5,4E-02
R-HSA-446728	Cell junction organization	16	126	0,0080	8,5E-04	5,8E-02
R-HSA-9013149	RAC1 GTPase cycle	21	191	0,0121	9,1E-04	5,9E-02
R-HSA-1236977	Endosomal/Vacuolar pathway	5	15	0,0009	1,1E-03	6,7E-02
R-HSA-983170	Antigen Presentation: Folding, assembly and peptide loading of class I MHC	8	41	0,0026	1,3E-03	7,7E-02
R-HSA-8957275	Post-translational protein phosphorylation	14	109	0,0069	1,6E-03	8,9E-02
R-HSA-5578775	Ion homeostasis	10	64	0,0040	1,8E-03	9,7E-02

Discussion

1. The authors suggest that avanafil might not work in the patients with MT-ATP6 mutations included in this manuscript due to the fact that their membrane potential was unaltered:

“This genotypic specificity may also explain why avanafil treatment, which successfully rescued an MT-ATP6 missense variant that reduced mitochondrial membrane potential, did not alleviate the hypermetabolic phenotype in our MT-ATP6 nonsense mutant neurons.”

I feel that this statement might be a bit preliminary. The authors solely test levels of glucose incorporation into lactate and citrate levels after avanafil treatment. It would be interesting to test the effects of avanafil on multiple aspects of mitochondrial functioning, including basal resp. capacity, proton leak, pantoate levels, etc. before definite conclusions can be drawn about how and why avanafil does not change lactate and citrate levels. I suggest that the authors either remove the aforementioned sentence or rephrase it to something along the lines of: “Avanafil did not alter levels of glucose incorporation into lactate or citrate, suggesting that this drug cannot change the metabolic changes observed in our MT-ATP6 nonsense mutant.”

We have re-analyzed our metabolomics data and modified Sup. Figure 3. We added now a new figure, Sup. Figure 3F, that shows pantothenate levels after avanafil treatment. A subset of these data was previously included in Sup. Fig. 3B, which included data from two additional independent metabolomics experiments, to validate the reduction of pantothenate in mutant motor neurons (only the untreated group was used). The new Sup. Fig. 3B depicts now a single independent metabolomics experiment to validate the pantothenate depletion. Following the new Sup. Figure 3F, pantothenate rises modestly after treatment in the 52 % heteroplasmy motor neurons, but the increase remains well below control levels and is not reproduced in the second biological replicate (54 % heteroplasmy). In this experiment, we also tested two lower avanafil concentrations (0.1 μ M and 0.5 μ M) with no effect (data not shown). Taken together with our citrate and lactate measurements after Avanafil treatment, and in line with Reviewer #3’s suggestion, we have revised the text. We removed the aforementioned sentence from the discussion, and we modified the results section as follows (204-209):

We next tested whether avanafil, a PDE5 inhibitor previously shown to rescue ATP synthase deficiency caused by an MT-ATP6 missense mutation, could restore the metabolic changes observed in mutant iPSC-MN. However, pantothenate consumption and ¹³C-glucose incorporation into lactate and citrate were unchanged in avanafil treated cells (Supplementary Fig. 3E), indicating that the drug failed to restore glucose utilization and TCA cycle activity under our experimental conditions.

Figure legends

1. **Figure 1:** For Figure 1G, three data points are shown for most cell lines, while for the 0% heteroplasmy (2) only one data point is shown – but still an error bar is present. Where is the error bar based on if there is only one data point? In addition, the authors state that for this experiment, $n = 3$ independent experiments with 2 to 11 wells per cell line per experiment were used. Could the authors specify in the figure legend what the dots represent? Different wells in one representative experiment? Averages?

Thank you for catching this. Each group does contain three data points, but the bar for the 0% (2) group was accidentally rendered in the same color as the individual data points. We have corrected the colors to eliminate this overlap and have standardized the grey-and-orange palette across all figures in the manuscript.

We measured mitochondrial membrane potential three times, using independent motor neuron differentiations (three differentiations, one measurement per differentiation). Only wells with a homogeneous monolayer of motor neurons were included in the analysis. Depending on the experiment, we obtained between 2 and 11 reliable replicate wells per cell line. For each differentiation, we calculated the average value across all wells for a given cell line, resulting in a single data point per differentiation. For the final graph, we included one data point per differentiation ($n = 3$ differentiations) and performed statistical analysis accordingly.

To make it clearer, we have modified the legend of Figure 1 and the Material and Methods section:

Figure 1 legend (lines 971-973): *Each dot represents an independent motor neuron differentiation experiment ($n_{exp} = 3$ independent differentiations) with an average of 2 to 11 wells per cell line per experiment.*

Material and Methods (lines 613-617): *Analyses were performed with Harmony 4.9 software using following standard pipelines. For each motor neuron differentiation ($n_{exp} = 3$ independent differentiations), the average value across all wells for each cell line was calculated ($n =$ between 2 and 11 wells), resulting in a single data point per differentiation*

2. Figure 3C: The Y-axis label “selected upregulated proteins” seems to be cropped incorrectly.

We missed this and will make sure to fix it, thank you.

3. The labelling of Figure 4 starts with (C). Should be (A)?

Again, thank you very much for spotting this. We will change it.

4. Supplementary Figure 1D. What do the dots represent? “The data represents $n=3$ independent differentiations experiments, each with two independent wells measured by duplicate.” The two independent wells or three independent experiments?

To make it clearer, we added this text to the legend of Supp. Figure 1D:

(D) *Quantification of (C) as the fraction of mutant peak intensity / total. Each dot represents an independent motor neuron differentiation ($n_{exp} = 3$ independent differentiations). For each differentiation, heteroplasmy levels were measured from two separate wells ($n_{technical_replicates} = 2$), with each measurement performed in duplicate. The average of the two technical replicates was used to generate a single data point for each differentiation*